# LENS: Multi-level Evaluation of Multimodal Reasoning with Large Language Models

**Ruilin Yao**[1,3], **Bo Zhang**[1], **Jirui Huang**[1,3], **Xinwei Long**[2], **Yifang Zhang**[1],
**Tianyu Zou**[1], **Shili Xiong**[1], **Yi Rong**[1], **Yufei Wu**[1], **Shichao Su**[1], **Yifan Xu**[1]
**Wenxi Zeng**[1], **Zhaoyu Yang**[1], **Guoyou Li**[1], **Shilan Zhang**[1], **Zichan Li**[1],
**Yaxiong Chen**[1], **Shengwu Xiong**[1,*] **Peng Xu**[2,*] **Jiajun Zhang**[3],
**Bowen Zhou**[2,4], **David A. Clifton**[5], **Luc Van Gool**[6]
[1] Wuhan University of Technology    [2] Tsinghua University
[3] Institute of Automation, Chinese Academy of Sciences [4] Shanghai AI Lab
[5] University of Oxford    [6] INSAIT, Sofia Un. St Kliment Ohridski
xiongsw@whut.edu.cn, peng_xu@tsinghua.edu.cn

## Abstract

Multimodal Large Language Models (MLLMs) have achieved significant advances in integrating visual and linguistic information, yet their ability to reason about complex and real-world scenarios remains limited. Existing benchmarks are usually constructed in a task-oriented manner, without a guarantee that different task samples come from the same data distribution. Therefore, they often fall short in evaluating the synergistic effects of lower-level perceptual capabilities on higher-order reasoning. To lift this limitation, we contribute Lens, a multi-level evaluation benchmark of multimodal reasoning with 3.4K contemporary images and 60K+ human-authored questions covering eight tasks and 12 daily scenarios, forming three progressive task tiers, *i.e.*, perception, understanding, and reasoning. One feature is that each image is equipped with rich annotations for all tasks. Thus, this data set intrinsically supports evaluating MLLMs to handle image-invariable prompts, from basic perception to compositional reasoning. In addition, our images have been collected manually from social media, with 53% published after Jan. 2025. We evaluate 15+ frontier MLLMs such as Qwen2.5-VL, InternVL3, GPT-4o and two reasoning models QVQ-Max and Kimi-VL. Most models were released in 2025, and none of them achieve an accuracy beyond 60% in the reasoning tasks. Furthermore, we propose the Self-Driven Multi-Expert Collaborative Framework (SMEC), a framework designed for MLLMs that simulates a panel of experts discussing and exchanging viewpoints via self-generated role-specific prompts. The experimental results confirm the existence of synergistic effects in a hierarchical task structure, where low-level tasks facilitate the reasoning of MLLMs on more complex, high-level tasks. Statistical analysis and ablation studies further demonstrate the comprehensiveness of our dataset and the superiority of our methodology. Project page: https://github.com/Lens4MLLMs/lens. We conducted the ICCV 2025 MARS2 Multimodal Reasoning Challenge on Lens https://mars2workshop.github.io/iccv2025/

## 1 Introduction

Multimodal Large Language Models (MLLMs) have emerged as a rapidly advancing field in artificial intelligence, demonstrating substantial improvements in visual content recognition and multimodal reasoning (Zhu et al., 2025; Bai et al., 2025; Wu et al., 2024; Team et al., 2024; 2025a). Despite their promising capabilities, MLLMs continue to face significant challenges in interpreting complex and real-world visual environments that are inherently dynamic, diverse, and grounded in physicality. However, existing benchmarks remain limited in their ability to evaluate multi-level reasoning.

---

*Corresponding authors.

Early evaluations were largely based on classical computer vision tasks (Everingham et al., 2010; Lin et al., 2014; Yu et al., 2016) and their integration with natural language. The real-world knowledge was often superficial, resulting in weak alignment between visual input and linguistic output. Secondly, these benchmarks are typically constructed under closed-world assumptions, lacking the inter-task consistency needed to assess reasoning across modalities (Fu et al., 2024; Li et al., 2024a). As a result, the absence of quantitative multi-level evaluation hinders meaningful comparison across MLLMs.

More recent benchmarks have begun to shift toward open-world evaluation and multimodal reasoning tasks (Wu & Xie, 2024; Zang et al., 2025). While this represents progress, current benchmarks do not adequately assess the nuanced performance necessary to evaluate MLLMs' progression towards human-like intelligence in real-world settings. They require largely primary visual comprehension and fall short of measuring higher-order reasoning and spatial understanding (Yue et al., 2024; Liu et al., 2024c). Furthermore, data distributions often differed between tasks, so that high performance in perceptual tasks did not necessarily translate into strong inference capabilities in more complex integrated multimodal tasks (Fan et al., 2025). As a result, they ignore the synergistic effect of the combinations of lower-order perceptual abilities on higher-order reasoning and are hard to provide a fine-grained assessment.

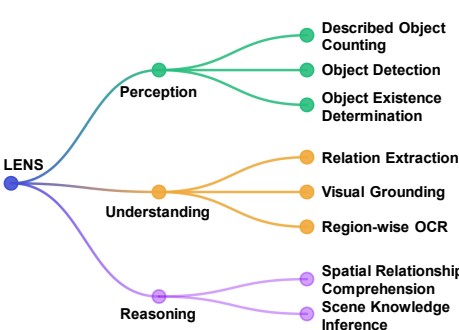

Figure 1: Illustration of the task split in `Lens`.

In this study, we propose a hierarchical and comprehensive evaluation framework `Lens` specifically designed to assess the multimodal capabilities in real-world scenarios. Our benchmark focuses on both isolated tasks and the integration of perception, understanding, and reasoning—three core tiers essential for intelligent multimodal systems. As shown in Figure 1, `Lens` encompasses eight tasks, systematically organized into three hierarchical tiers with eight subtasks, and it comprises 3.4K real-world photographs and 60K+ human-authored questions, in 12 diverse scenarios—including streets, stations, schools, homes, and more, which can be roughly divided into three themes: "Home", "Education", and "City", and we visualize the high-frequency words under different themes in Figure 2. 53% of the images are from 2025 and more than 80% of the images are from after September 2024, ensuring the content reflects contemporary environments.

For task design, `Lens` adopts an open-set configuration, allowing queries to be posed in natural language and grounded in authentic photographic content. This design enables evaluation of model performance in complex, ambiguous, and information-rich settings, better aligning with real-time human demands. Moreover, our benchmark introduces multi-level tasks, which are unified by shared visual contexts, making `Lens` well-suited for assessing the synergistic effects of lower-level perceptual abilities (*e.g.*, object detection, localization) on higher-order reasoning tasks. To succeed in `Lens`, models must jointly process multimodal input, recall domain knowledge, and conduct multi-step reasoning to arrive at valid conclusions. Our experimental results confirm that current state-of-the-art MLLMs still struggle with these reasoning-heavy tasks, revealing a significant gap between perception and functional understanding.

To bridge this gap, we propose the Self-Driven Multi-Expert Collaborative Framework (SMEC), a novel reasoning framework that leverages the MLLM itself as a set of specialized experts instantiated through self-generated prompts. Unlike tool-calling approaches (Wang et al., 2025; Gao et al., 2025; Liu et al., 2025b; Zhang et al., 2024b) that rely on external modules, SMEC treats the base MLLM as a versatile reasoning engine: it simulates diverse expert perspectives (*e.g.*, spatial analyst, text interpreter, commonsense reasoner) via role-specific prompts and composes their insights into coherent final answers. This collaborative mechanism encourages the model to extract, expand, and integrate rich, task-relevant information. Our experiments demonstrate that SMEC significantly boosts performance on reasoning tasks within `Lens`, validating its potential as a general-purpose, language-native method for enhancing multimodal reasoning. **We take data privacy, copyright compliance, and platform terms of service seriously. Rigorous collection, filtering, and documentation procedures were implemented and detailed in Section 2.1 and 6, and Appendix A.4.** To our knowledge, we make the following contributions:

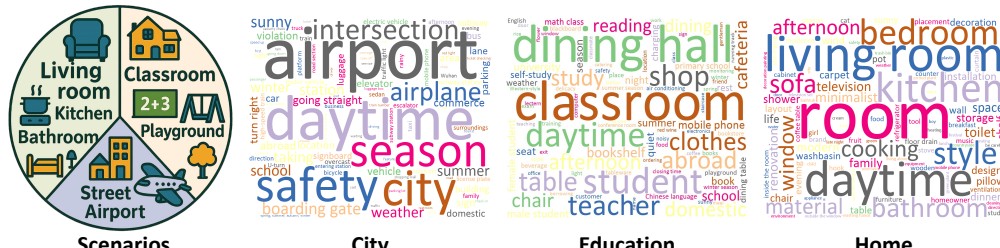

Figure 2: Three core themes, "Education", "City", and "Home", along with their word clouds of the scenario distributions by name size.

- **Realistic and Up-to-Date Evaluation.** By leveraging a newly collected set of high-resolution, naturalistic images, our benchmark evaluates the latest multimodal reasoning models in settings that closely reflect real-world complexity.

- **Multi-Level Evaluation.** It supports fine-grained and interpretable evaluation across three core dimensions—perception, understanding, and reasoning—providing a comprehensive view of a model's multimodal competence.

- **Synergistic Capability Evaluation.** Unlike existing benchmarks that often assess tasks in isolation, our framework emphasizes the synergistic effects of lower-level perceptual abilities on higher-order reasoning tasks. The experimental results also confirm that low-level tasks facilitate the reasoning of MLLMs on more complex, high-level tasks (*e.g.*, Scene Knowledge Inference).

- **Towards Generalizable Intelligence.** By capturing both perceptual and reasoning performance in integrated tasks, our benchmark helps identify the gaps between current model capabilities and the requirements of human-aligned reasoning systems and measure the shortcomings of current models.

- **Self-driven Reasoning Enhancement.** We introduce SMEC, a self-driven multi-expert collaborative framework that simulates specialized experts within a single MLLM through self-generated prompts. Unlike tool-calling approaches, SMEC enables modular, multi-perspective reasoning natively, leading to significant gains on complex reasoning tasks.

**Comparison with existing benchmarks.** Compared with existing multimodal benchmarks (Liu et al., 2024c; Yue et al., 2024; Li et al., 2024b), `Lens` provides more contemporary, diverse, and densely annotated visual content. Our benchmark is constructed from contemporary social-media images, ensuring strong timeliness and significantly reducing the risk of contamination from pre-training corpora. In contrast to task-specific datasets (Liu et al., 2024d;a; Wei et al., 2024), our benchmark provides rich, multi-task annotations with the same visual content, across perception, understanding, and reasoning, enabling controlled analysis of cross-task synergies within a unified distribution. Additionally, `Lens` offers the detailed thought process in real-world reasoning tasks for potential future research. Appendix A.1 further discusses related work.

## 2 LENS DATASET AND BENCHMARK

### 2.1 DATA COLLECTION

The image data collection in our benchmark focuses on real-world scenes to ensure diversity, representativeness, and practicality for visual perception, understanding, and reasoning tasks. To this end, we first defined a set of common real-life scenarios that are highly relevant to typical human visual experiences. The selection principle was that each visual scene should contain distinguishable and representative semantic content. For example, street scenes are usually populated with cars, pedestrians, and storefronts, while indoor environments like classrooms often involve students, teachers, and educational materials. To avoid regional or cultural bias and ensure a broad distribution of content, we collected images from multiple social media platforms, including X (formerly Twitter), Instagram, Weibo, and RedNote. These platforms were chosen due to their global user bases and diverse content coverage across regions and lifestyles. During the collection process, we strictly complied with the copyright and licensing regulations of each platform, ensuring that data was collected only from

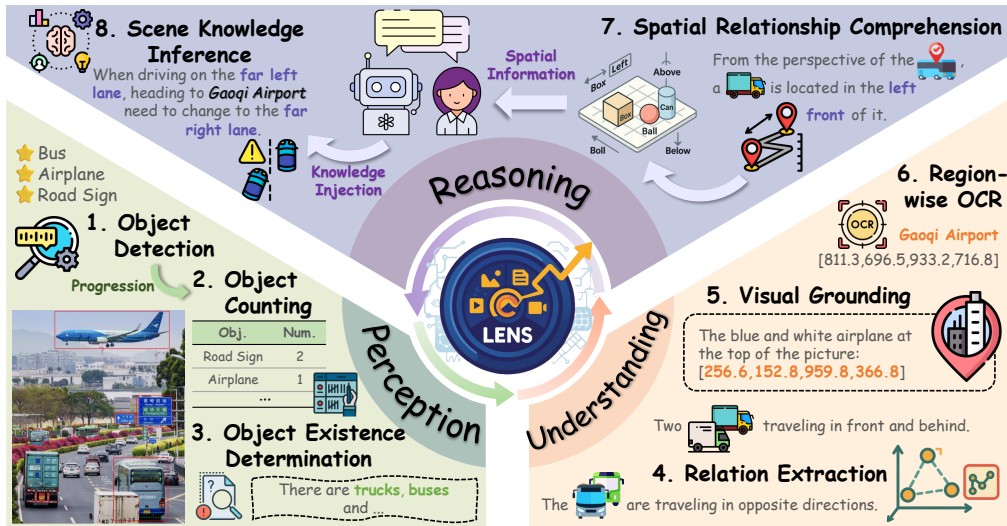

Figure 3: `Lens` consists of eight tasks at three levels. **Perception** tasks focus on recognizing object attribute and counting. **Understanding** tasks emphasizes localization and inter-object relationships with textual information. **Reasoning** tasks demand the use of external knowledge beyond the visual input and involve multi-step, complex reasoning processes to arrive at the correct answer.

Table 1: Comparison with other recently released multimodal benchmarks.

| Benchmarks | Venue | Att. | Cnt | Loc | Rel | Reasoning | Interleaved Image-Text | Image Source |
|---|---|---|---|---|---|---|---|---|
| V* (Wu & Xie, 2024) | CVPR'24 | ✗ | ✗ | ✔ | ✔ | ✔ | ✗ | SA-1B (Kirillov et al., 2023) |
| SPEC (Peng et al., 2024) | CVPR'24 | ✔ | ✔ | ✔ | ✔ | ✗ | ✗ | Synthesize |
| MMVP (Tong et al., 2024) | CVPR'24 | ✔ | ✗ | ✗ | ✗ | ✗ | ✗ | ImageNet (Russakovsky et al., 2015), LAION-5B (Schuhmann et al., 2022) |
| HaloQuest (Wang et al., 2024b) | ECCV'24 | ✔ | ✗ | ✗ | ✗ | ✔ | ✗ | Open Images (Kuznetsova et al., 2020) |
| AS-V2 (Wang et al., 2024a) | ECCV'24 | ✔ | ✔ | ✔ | ✔ | ✔ | ✗ | COCO (Caesar et al., 2018) |
| MMBench (Liu et al., 2024c) | ECCV'24 | ✔ | ✔ | ✔ | ✔ | ✔ | ✗ | Internet images |
| HC-RefLoCo (Wei et al., 2024) | NeurIPS'24 | ✗ | ✗ | ✔ | ✔ | ✔ | ✗ | Multiple existing datasets |
| Visual CoT (Shao et al., 2024) | NeurIPS'24 | ✔ | ✗ | ✔ | ✗ | ✔ | ✗ | Multiple existing datasets |
| MC-Bench (Xu et al., 2024) | arXiv'24 | ✗ | ✗ | ✔ | ✗ | ✔ | ✔ | Multiple existing datasets, Internet |
| CODE (Zang et al., 2025) | IJCV'25 | ✔ | ✔ | ✔ | ✗ | ✗ | ✗ | Flickr30k series (Young et al., 2014; Plummer et al., 2015) |
| ChatterBox (Tian et al., 2025) | AAAI'25 | ✔ | ✔ | ✔ | ✗ | ✔ | ✗ | Visual Genome (Krishna et al., 2017) |
| `Lens` | - | ✔ | ✔ | ✔ | ✔ | ✔ | ✔ | Collect manually from social media 53% published later than Jan. 2025 |

"Att.": Attribute; "Cnt": Count; "Loc": Localization; "Rel": Relation

publicly accessible posts and that no images were downloaded from sources explicitly prohibiting data reuse or redistribution. Moreover, to facilitate the evaluation of multiple subtasks within the same image (*e.g.*, detection, OCR, scene knowledge inference), we curated images that exhibit rich semantic content while maintaining scene clarity. Complex or ambiguous images were manually filtered out to avoid introducing noise that could hinder benchmarking or evaluation consistency. Please note that we manually collect these data that are completely open to the Internet and have complied with the developer agreement of the relevant platform (*e.g.*, Developer Policy of X and Meta ), ensuring non-commercial use, erasing geographic information, user personal information, etc. from the original data. For further details, please refer to the appendix A.4.

## 2.2 TASK DESIGN AND ANNOTATION PROCESS

To construct a comprehensive and diverse benchmark, we recruited over 50 undergraduate and graduate students (including authors) as human annotators to assist in the process of question collection and task annotation and paid the corresponding salary. These annotators were carefully trained to ensure high annotation quality and consistency. As shown in Figure 3, the generated questions were divided into three major categories: Perception, Understanding, and Reasoning. For Perception and Understanding, they primarily target the model's ability to perceive visual objects and align them accurately with natural language descriptions. They emphasize fine-grained

---

https://developer.x.com/en/use-cases/do-research/academic-research
https://developers.facebook.com/docs/instagram-platform

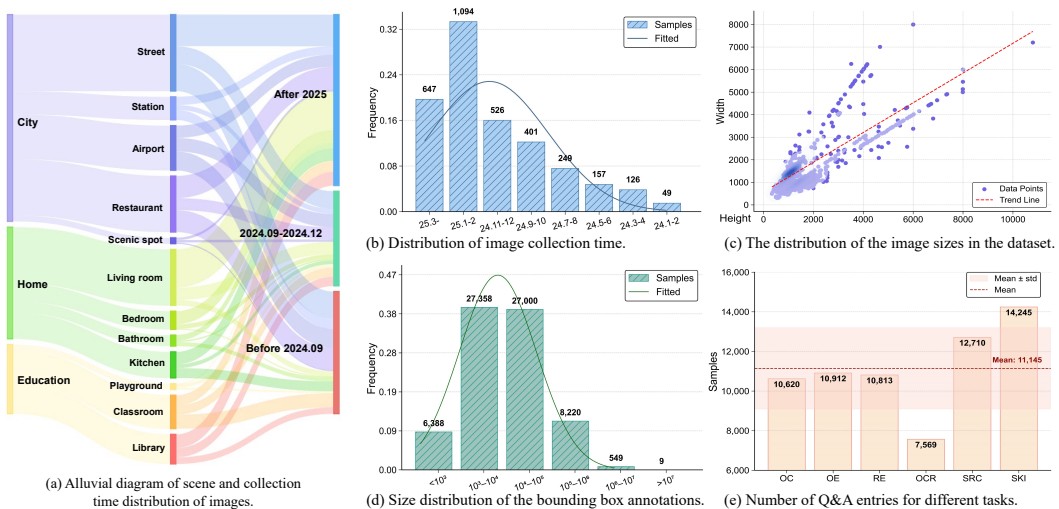

Figure 4: Statistical analysis of our dataset. We visualize the temporal distribution of the images for different scenarios, size distribution of images and bounding box annotations, and number of QA entries for different tasks, demonstrating the timeliness and diversity of our data.

visual grounding and object recognition rather than abstract reasoning. At last, reasoning-based questions aim to evaluate the model's ability to understand user intent and reason based on external knowledge, commonsense, physical laws, or background information beyond the purely visual content of the image. Based on these assessment dimensions, we compare `Lens` with related multimodal benchmarks in Table 1 and formulate our challenging open-ended, language-driven tasks: Object Counting (OC), Object Detection (OD), Object Existence Determination (OE), Relation Extraction (RE), Visual Grounding (VG), Region-wise OCR (OCR), Spatial Relationship Comprehension (SRC), and Scene Knowledge Inference (SKI). We provide a more detailed introduction of these tasks in Appendix A.2.

## 2.3 DATA ANALYSIS

We aim to construct a dataset that is not only comprehensive and dynamic but also emphasizes reasoning capabilities. In the following analysis, we demonstrate the strengths of our benchmark in terms of diversity of images and annotations. The quantitative results are visualized in Figure 4.

First, our benchmark incorporates scene-aware content and real-time data. As shown in Figure 4, more than 50% of the images in our dataset were collected in 2025, and approximately 70% were collected in November 2024 and beyond, which avoids potential data leakage. Many images reflect dynamic scenes (*e.g.*, crowded streets, interactive environments) captured at different times and locations, aligning with real-world scenarios.

Second, in our dataset, the coverage of a wide range of object categories, scene types, and bounding box annotations further supports diverse downstream tasks from detection to high-level semantic inference and interleaved image-text understanding. As illustrated in Figure 4 (c), the high resolution of the images in our dataset makes it challenging for fine-grained understanding of the model and supports evaluation across varying input sizes. Additionally, as shown in Figure 4 (d), the various objects are labeled with different sizes of bounding boxes to meet the needs of multi-scale object detection and region-wise OCR evaluation.

Furthermore, beyond perception, our dataset facilitates reasoning-oriented research by supporting tasks that require: Spatial reasoning (*e.g.*, understanding object layouts and geometric relationships). Relational inference (*e.g.*, extracting interactions between objects). Commonsense knowledge application (*e.g.*, inferring the feasibility of a behavior or scene functionalities). Cross-modal alignment (*e.g.*, grounding free-form language to specific visual content). We also analyze the question-answer pairs distribution of these tasks and Figure 4 (e) shows that over 60% of the questions in the dataset go beyond simple recognition, explicitly encouraging models to reason

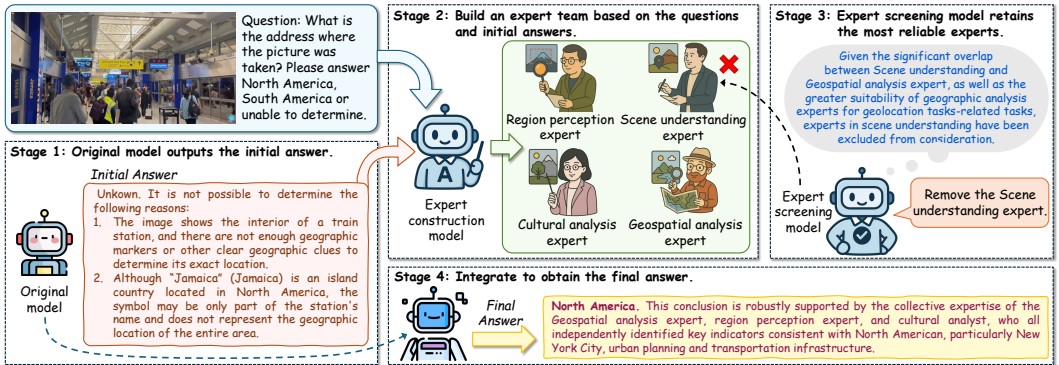

Figure 5: SMEC is a step-wise reasoning framework for answering complex visual questions using the Multi-Expert Collaboration. Starting from an initial model response, SMEC constructs a specialized expert team, to re-evaluate and refine the answer. Through expert screening and integration, unreliable and duplicated experts are filtered out, and a consensus-based final answer is produced, demonstrating the advantages of modular, expert-driven collaboration in visual reasoning tasks.

about the scene, context, and user intent. Please refer to the Appendix A.5 for more low-level visual analysis.

## 3 SELF-DRIVEN MULTI-EXPERT COLLABORATIVE FRAMEWORK

We propose a Self-Driven Multi-Expert Collaborative Framework to tackle complex visual reasoning tasks that require diverse domain expertise and multi-level inference. Built on an instruction-tuned MLLM, our framework dynamically assembles a set of self-generated experts, each embodying a distinct reasoning perspective, as shown in Figure 5.

**Expert Generation via Prompted Role Construction.** Given a query $q$, the base model $\theta$ first produces a coarse initial answer $a_0$. To enrich the reasoning space, a Meta Generation Prompt $p_g$ is used to iteratively generate expert role descriptions $d_q^t$, simulating specialized agents (*e.g.*, geospatial analyst, cultural analyst). Each valid description yields a new expert response $a_t$, which is added to the answer set $A$. This loop continues until either a diversity criterion is violated or a maximum number of iterations $N_t$ is reached.

**Prompt Adaptation and Redundancy Filtering.** To avoid degenerate expert generation, the framework dynamically updates $p_g$ when semantically redundant descriptions emerge. This adaptation encourages exploration of novel expert roles. Implicit expert screening is performed by discarding repetitive or low-information descriptions, ensuring a concise yet diverse expert team with minimal computational overhead.

**Consensus-Driven Answer Integration.** The final stage aggregates the expert responses via a Collaboration Prompt $p_c$, prompting $\theta$ to synthesize a unified answer $a_{\text{final}}$ through deliberative reasoning. This mimics human expert panels that reconcile differing views to reach a robust consensus.

We detail a formal description of the process in Appendix A.10. In this way, our framework instantiates modular experts purely via prompt-based self-conditioning. Unlike fixed-rule multi-agent systems and tool-calling methods, our framework leverages the generative flexibility of MLLM to dynamically instantiate and evolve its behaviors, without requiring external task-specific supervision.

## 4 EXPERIMENTS

### 4.1 EVALUATION MODELS

To illustrate the difficulty of our benchmark and evaluate the latest advances in current research, we evaluate various MLLMs belonging to three major categories: Closed-source generalist MLLMs, such as GPT-4o (Achiam et al., 2023) and Gemini2.5 Pro (Team et al., 2024). Open-source generalist MLLMs like Qwen2.5-VL (Bai et al., 2025), Deepseek-VL2 (Wu et al., 2024), Gemma3 (Team et al.,

Table 2: Comparison of state-of-the-art methods on `Lens`. We evaluate object detection (OD) performance using $AP_{50}$ (Lin et al., 2014), visual grounding (VG) performance with ACC@0.5 (Xiao et al., 2024), and use accuracy for other tasks. Task abbreviations follow the definitions provided in Section 2.2. "MoE 1B/3B" denotes 3B Mixture of Experts model with 1B parameters activated. "N/A" denotes the official documentation does not confirm that the model is applicable for the task. Best performing models are shaded in red.

| Methods | Model size | Perception | | | Understanding | | | Reasoning | |
|---|---|---|---|---|---|---|---|---|---|
| | | OC | OD | OE | RE | VG | OCR | SRC | SKI |
| **MLLM (closed source)** | | | | | | | | | |
| GPT-4o | - | 54.32 | N/A | 85.09 | 72.77 | N/A | 42.86 | 51.14 | 55.20 |
| Gemini2.5-Pro | - | 60.18 | 47.40 | 86.59 | 76.52 | 25.61 | 61.95 | 56.20 | 59.31 |
| **Open source** | | | | | | | | | |
| Deepseek-VL2-tiny | MoE 1B/3B | 56.22 | 21.12 | 72.11 | 58.73 | 16.09 | 44.01 | 38.97 | 45.12 |
| Deepseek-VL2 | MoE 4.5B/27B | 61.41 | 46.08 | 77.68 | 69.18 | 42.47 | 48.76 | 44.58 | 49.50 |
| Gemma3 | 4B | 38.85 | N/A | 71.88 | 62.98 | N/A | 27.03 | 39.53 | 45.18 |
| Gemma3 | 12B | 44.65 | N/A | 73.21 | 62.78 | N/A | 33.98 | 43.33 | 48.56 |
| InternVL3 | 2B | 55.81 | 18.39 | 71.96 | 64.49 | 15.22 | 45.51 | 40.56 | 48.59 |
| InternVL3 | 9B | 55.63 | 25.79 | 77.49 | 67.18 | 18.18 | 48.79 | 44.69 | 51.32 |
| InternVL3 | 38B | 62.78 | 43.44 | 81.60 | 71.37 | 24.98 | 51.72 | 47.18 | 51.85 |
| InternVL3 | 78B | 61.38 | 47.44 | 84.87 | 74.93 | 27.24 | 54.21 | 49.39 | 55.17 |
| Qwen2.5-VL | 3B | 58.76 | 35.16 | 74.01 | 66.52 | 39.44 | 52.43 | 40.33 | 46.50 |
| Qwen2.5-VL | 7B | 58.35 | 37.75 | 83.75 | 71.58 | 40.11 | 61.65 | 46.28 | 48.87 |
| Qwen2.5-VL | 32B | 62.25 | 39.93 | 83.60 | 74.57 | 41.15 | 65.64 | 51.66 | 51.54 |
| Qwen2.5-VL | 72B | 59.75 | 43.48 | 85.67 | 75.98 | 44.98 | 68.51 | 53.65 | 54.79 |
| **Reasoning model** | | | | | | | | | |
| QVQ-Max | 72B | 49.95 | N/A | 85.37 | 74.01 | N/A | 58.67 | 50.80 | 58.86 |
| Kimi-VL-thinking | MoE 2.8B/16B | 46.87 | N/A | 72.77 | 48.16 | N/A | 30.21 | 29.40 | 36.44 |

2025a), InternVL3 (Zhu et al., 2025). Multimodal reasoning models QvQ-Max, Kimi-VL-thinking (Team et al., 2025b) and GLM-4.1V-Thinking (Hong et al., 2025), focusing on advanced reasoning capabilities. The release dates of these models are distributed from Dec. 2024 to Apr. 2025.

## 4.2 EVALUATION STRATEGY

To ensure a fair and efficient assessment of model performance across our benchmark, we adopt two evaluation strategies for main results. For perception and understanding tasks, models were evaluated based on their direct outputs without additional inference-time computations. For complex reasoning tasks, which require deeper multi-step inference, we allow models to generate multiple candidate responses per question and the final prediction is then selected via majority voting (Liu et al., 2025a). For qualitative judgment, we follow prior work (Wang et al., 2023) and employ a large language model (*e.g.*, GLM4-flash (GLM et al., 2024)) as an automatic evaluator. The LLM is prompted to produce multiple pieces of evaluation evidence for calibration, comparing the model-generated responses against human-annotated answers, aiming to offer a consistent framework for evaluating model performance across diverse tasks.

## 4.3 EVALUATION RESULTS

We evaluate a suite of state-of-the-art Multimodal Large Language Models on our benchmark, which spans three tiers and eight tasks. Results, as shown in Table 2, reveal insights into model scaling, inter-task dependencies, and capability gaps in current MLLMs.

**Model Scaling and General Trends.** We observe a consistent performance gain with increased model size in both closed- and open-source models. For example, Qwen2.5-VL improves steadily from 3B to 72B, achieving top performance on reasoning tasks. InternVL3 shows similar gains in

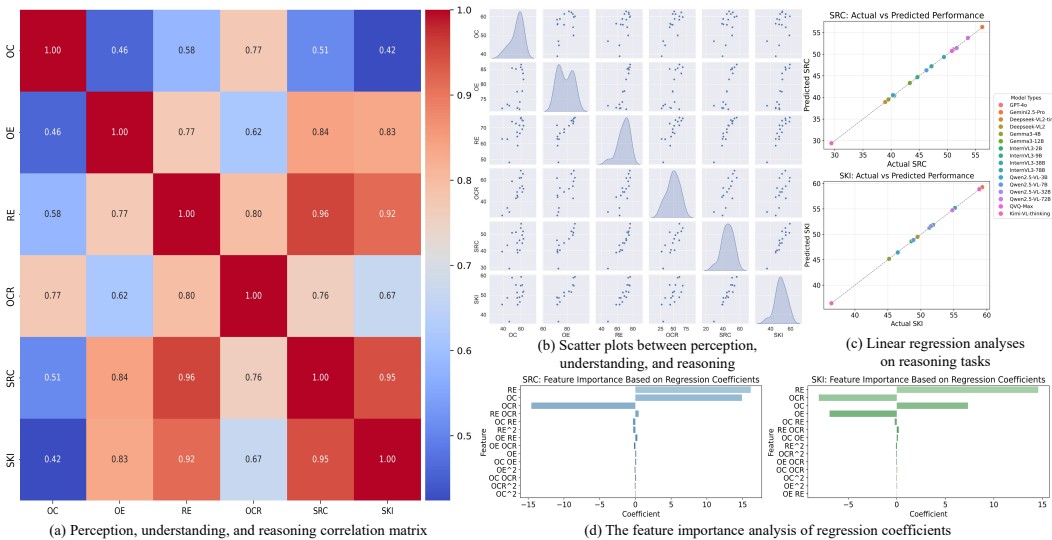

Figure 6: Statistical analysis of model accuracy and synergies between different tasks.

OD, rising from 18.39% (2B) to 47.44% (78B), though performance saturates at higher scales. These trends confirm that scaling remains a key driver for multimodal reasoning, albeit with diminishing returns in some subtasks.

**Perception: Foundation for Higher Cognition.** Perception-level tasks form the backbone of visual reasoning. Closed-source models like Gemini2.5-Pro and GPT-4o excel at OE (86.59% and 85.09%, respectively), although OD support is lacking. Among open-source models, Deepseek-VL2 and Qwen2.5-VL-72B deliver competitive OD and OC performance. Notably, models with stronger perception capabilities tend to exhibit superior reasoning performance, highlighting the foundational role of low-level visual understanding.

**Understanding: Progress and Bottlenecks.** Understanding tasks assess models' ability to interpret structured visual semantics with textual information. Gemini2.5-Pro leads in RE (76.52%) and OCR (61.95%), showcasing robust relational and textual grounding. However, VG remains a bottleneck even for large-scale models like InternVL3-78B (27.24%) and Qwen2.5-VL-72B (44.98%), suggesting persistent challenges in fine-grained spatial-semantic alignment.

**Reasoning: High-Level Generalization.** Reasoning tasks are the most demanding. Closed-source models such as GPT-4o and Gemini2.5-Pro achieve strong results (51.14%/56.20% on SRC and 55.20%/59.31% on SKI). Among open-source models, Qwen2.5-VL-72B leads, while the reasoning-specialized QVQ-Max approaches closed-source performance (58.86% on SKI) despite lacking OD and VG capabilities. This suggests that explicit reasoning models can partially compensate for perceptual limitations, likely relying on test-time scaling rather than grounded perception.

## 4.4 SYNERGISTIC EFFECTS ANALYSIS

To analyze the cross-task performance patterns of different models, we perform a statistical analysis of the synergies between different tasks and visualized the results as in Figure 6. We compute the Pearson correlation coefficients between Perception and Understanding tasks and observe notable interdependencies. OC and RE exhibit a strong positive correlation of 0.73, while OE and OCR show a similarly significant correlation of 0.67. These results indicate that effective performance in perception directly contributes to understanding, which in turn underpins downstream reasoning. Scatter plot visualizations further confirm these links, OCR, in particular, correlates strongly with both SRC and SKI, underscoring its central role in enabling semantic reasoning. Linear regression analyses reinforce these findings: OE and OCR are strong predictors of SRC, while OC and RE significantly influence SKI, highlighting how object-level detection and relational reasoning jointly support high-level inference. Finally, we apply second-order polynomial regression and the feature importance analysis of regression coefficients reveals task-specific contributions. These insights

collectively demonstrate the layered structure of visual reasoning pipelines, where perception and understanding stages must be well-aligned to support robust inference. For further analysis, please refer to the appendix A.7.

## 4.5 EFFECTIVENESS OF SMEC

To further evaluate the effectiveness of our proposed SMEC framework, we conduct experiments on the Scene Knowledge Inference (SKI) task using both a sampled subset of 3,500 question–answer pairs and the full test set (Table 3). Compared to the direct inference baseline, SMEC consistently improves performance across both model scales and different iteration depths. For Qwen2.5VL-7B, accuracy increases from 39.80% to 43.24% as the number of iterations $N_t$ grows from 1 to 3, demonstrating the benefits of multi-step expert collaboration. Notably, SMEC also outperforms Self-Refine

Table 3: Accuracy comparison with different settings.

| Methods | Model | Iterations | Performance |
|---|---|---|---|
| Direct | Qwen2.5VL-7b | - | 39.80 |
| Majority voting | Qwen2.5VL-7b | - | 40.66 |
| Self-Refine | Qwen2.5VL-7b | - | 40.51 (+0.71) |
| SMEC | Qwen2.5VL-7b | 1 | 41.35 (+1.55) |
| SMEC | Qwen2.5VL-7b | 2 | 42.97 (+3.17) |
| SMEC | Qwen2.5VL-7b | 3 | 43.24 (+3.44) |
| Direct | Qwen2.5VL-32b | - | 49.17 |
| SMEC | Qwen2.5VL-32b | 3 | 52.44 (+3.27) |
| Direct (Full data) | Qwen2.5VL-32b | - | 51.54 |
| SMEC (Full data) | Qwen2.5VL-32b | 3 | 54.66 (+3.12) |

(Madaan et al., 2023) and Majority voting (Chen et al., 2024b) under the same setting. A similar trend appears at larger scales. With Qwen2.5VL-32B, SMEC improves accuracy from 49.17% to 52.44% with three iterations, confirming that the benefits of iterative expert collaboration scale with model capacity. Importantly, when evaluated on the full test set rather than the 3,500-sample subset, SMEC continues to provide consistent improvements (+3.12%). This indicates that SMEC's gains are not distribution-specific and remain stable under more comprehensive evaluation conditions.

## 5 CONCLUSION

We contribute `Lens`, a multi-level benchmark designed to evaluate Multimodal Large Language Models (MLLMs) across perception, understanding, and reasoning. Unlike prior benchmarks, `Lens` aligns all tasks to the same set of realistic, contemporary images, enabling fine-grained analysis of how low-level visual capabilities support higher-order reasoning. The evaluation of recent MLLMs further reveals a consistent performance gap in reasoning tasks, highlighting the limitations of current models in integrating perception and cognition. To address this, we proposed SMEC, a self-driven multi-expert collaborative framework that prompts the MLLM to simulate a panel of specialized agents. Together, `Lens` and SMEC offer a new paradigm for evaluating and enhancing reasoning intelligence in MLLMs, paving the way to more robust, human-aligned multimodal intelligence.

## 6 ETHICS STATEMENT

**Data Collection and Privacy.** All images in the `Lens` dataset were **collected manually from publicly available posts** on social media platforms. We respect and strictly complied with the developer agreements and copyright regulations of these platforms. No private or restricted data was accessed, and all collection adhered to academic research policies. This dataset is intended strictly for non-commercial academic research purposes. If any content in this project is found to raise concerns related to privacy, copyright, or legal compliance, please contact us at **yaoruilin@whut.edu.cn**. We will promptly review the request and are willing to remove, modify, or withdraw the sensitive data or materials.

**Human Annotation.** The dataset was constructed with the assistance of more than 50 undergraduate and graduate student annotators, who were trained to ensure annotation quality and consistency. All annotators were fairly compensated for their work. The annotation process involved only task-related labeling (*e.g.*, bounding boxes, question-answer generation) and did not involve collection of personal or sensitive information about the annotators.

**Bias, Fairness, and Representation.** Although images were sourced from global platforms to ensure diversity of cultural and regional content, dataset bias may still exist due to platform-level demographic imbalances and uneven scenario representation. We acknowledge that these limitations

may influence model evaluation results, and we encourage future work to further broaden geographic and cultural coverage.

**Research Integrity and Transparency.** This study follows established standards of research integrity. We provide detailed dataset descriptions, evaluation protocols, and implementation details. We have no conflicts of interest or external sponsorship that might bias the study.

## 7 REPRODUCIBILITY STATEMENT

We have taken extensive measures to ensure the reproducibility of our results. A complete description of the `Lens` dataset, including data collection principles, filtering steps, and annotation procedures, is provided in Section 2 and Appendix. Details of the evaluation tasks, metrics, and benchmark comparisons are reported in Section 4, with additional analyses in Appendix. The implementation of our proposed Self-Driven Multi-Expert Collaborative Framework (SMEC), including the expert construction, screening, and integration process, is fully described in Section 3 and formalized in Appendix A.9. Hyperparameter settings, model configurations, and ablation experiments are included in the appendix to facilitate replication of results. Together, these resources ensure that both dataset construction and methodological contributions can be faithfully reproduced by the research community.

## ACKNOWLEDGEMENT

This work is supported by the National Key Research and Development Program of China No.2022ZD0160604, and NSFC No.62306162.

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

## A APPENDIX

### A.1 RELATED WORK

#### A.1.1 BENCHMARKS FOR VISUAL CAPABILITY OF MLLMs

The capability of Visual Perception, Understanding and Reasoning is a foundational aspect of understanding benchmarks, which involves the ability to recognize and localize multiple objects, interpret various visual elements with complex emotional or implicit cues and summarize visual information for feedback and decision making (Li et al., 2024b). Specifically, Perception in MLLMs involves the classification, detection of basic visual objects (*e.g.*, dog, cat) and attributes (*e.g.*, color, lighting). These low-level perceptual capabilities are crucial for various applications, including recognition systems (Zhao et al., 2024) and visual quality enhancement (Zhang et al., 2024c). Understanding represents a sophisticated level of image understanding that focuses on the detailed and nuanced aspects of visual content. It includes recognizing and interpreting the visual-linguistic concepts, such as text recognition (OCRBench (Liu et al., 2024d)), Visual Grounding (RefCOCO (Yu et al., 2016), FineCops-Ref (Liu et al., 2024a), HC-RefLoCo (Wei et al., 2024)) and Referring Expression Generation (Visual Genome) (Krishna et al., 2017), which refers to the model's ability to accurately link visual elements with corresponding textual descriptions. Although tasks at this level begin to involve visual and textual alignment, they still do not require reasoning or external knowledge. For higher-order capability, reasoning in MLLMs involves advanced event understanding and deep meaning extraction from multimodal data. These capabilities include interpreting and responding to complex emotional cues across multiple modalities (Cheng et al., 2024), deriving subtle implicit meanings from visual and contextual information (Liu et al., 2023a), and a range of other competencies, including knowledge acquisition, language generation, spatial awareness, and cultural context integration (Rachabatuni et al., 2024).

#### A.1.2 REASONING CAPABILITY OF MLLMs

MLLMs have demonstrated remarkable reasoning capabilities, largely facilitated by test-time scaling (Dong et al., 2022; Wei et al., 2022), which allows feeding prompted samples and context. This capability has been further enhanced by chain-of-thought (CoT) prompting (Wei et al., 2022), which enables LLMs to generate coherent intermediate reasoning steps toward the final answer. Previous studies have shown that LLMs benefit from manually written demonstrations as well as zero-shot prompting outputs. However, due to the domain gap between various modalities, the current reasoning capability of MLLMs in the complex real-world environment is still limited. To address this limitation, researchers have focused on enhancing the reasoning capability of MLLMs in both the training and prompting paradigms. Flamingo (Alayrac et al., 2022) bridges the gap between these two modalities by pre-training on interleaved visual and textual data. Some other works, such as Shikra (Chen et al., 2023b) and Ferret (You et al., 2023), leverage visual grounding data (Xiao et al., 2024; Yao et al., 2024) to achieve fine-grained vision-language alignment. Furthermore, recent studies have also demonstrated that augmenting computing resources during the testing phase (test-time scaling) can enhance the reasoning capabilities of LLMs (Jaech et al., 2024). More specifically, Prompt-based Reasoning Meta-Systems (PRMS) can be employed to guide LLMs in evaluating and filtering intermediate "thinking" processes (Snell et al., 2024). This encourages the generation of more sophisticated reasoning steps during testing, ultimately leading to improved reasoning accuracy. Beyond that, some methods employ the external knowledge to focus on important visual details, like V* (Wu & Xie, 2024), Marvel (Jiang et al., 2024), and ICAL (Sarch et al., 2024), collecting a series of visual reasoning steps as training data. More recently, with the emergence of DeepSeek-R1 (Guo et al., 2025) demonstrating strong potential in LLM reasoning, research efforts have begun to explore reasoning-centric models and R1-style reinforcement learning strategies for understanding complex visual scenes and tasks. These studies (Huang et al., 2025; Shen et al., 2025a; Liu et al., 2025c) particularly emphasize the long-chain reasoning capabilities within MLLMs, aiming to enhance their performance in handling intricate visual-linguistic reasoning challenges.

#### A.1.3 LLM-BASED AGENTIC REASONING

The rapid progress of large language models (Achiam et al., 2023; Bai et al., 2025) has sparked significant interest in building autonomous agents capable of solving complex, multi-step reasoning

tasks. Leveraging the strong chain-of-thought (CoT) abilities of modern LLMs (Wei et al., 2022), these systems typically decompose a complex problem into a sequence of structured subtasks, invoke intermediate deliberation, and integrate the resulting insights to produce a final answer (Gupta & Kembhavi, 2023; Chen et al., 2023a). Recent developments in LLM-based autonomous agents highlight the importance of planning (Huang et al., 2024; Zhang et al., 2024a), tool usage (Yuan et al., 2025), memory (Zhang et al., 2025c), and persona (Chen et al., 2024a). In parallel, multi-agent frameworks such as MetaGPT (Hong et al., 2023), AgentVerse (Chen et al., 2023c), etc., demonstrate strong performance by orchestrating multiple interacting agents, often instantiated as distinct roles with specialized responsibilities. Despite their success, these systems rely heavily on manually designed personas, fixed role hierarchies, or hand-crafted coordination rules, which limits their flexibility and generalization across tasks and domains. Moreover, the dependence on external scaffolding or pre-specified agent behaviors often restricts the model's ability to adaptively adjust its internal reasoning pathway. To address these limitations, SMEC introduces a self-driven agentic reasoning mechanism that automatically generates diverse experts, elicits their reasoning, filters redundant experts, and synthesizes their perspectives into a final consensus, all within the model's own language-native inference loop.

## A.2 SPECIFIC DEFINATION OF DIFFERENT TASKS

**Object Counting (OC):** Estimating the number of object instances described by a free-form expression, often under complex conditions like occlusion, scale variation, or clutter.

**Object Detection (OD):** Localizing objects within an image by generating bounding boxes paired with corresponding class labels. In order to better match the real-life scenarios and practical applications, we construct more than 500 fine-grained object categories based on natural language.

**Object Existence Determination (OE):** Determining whether a particular object, which described by a detailed expression, exists in the image without requiring spatial localization.

**Relation Extraction (RE):** Identifying semantic relationships (*e.g.*, "holding", "next to", "wearing") between pairs of objects to facilitate structured scene understanding. And we added questions about the objects that do not exist in the images to evaluate model's ability to suppress hallucinations.

**Visual Grounding (VG):** Localizing an image region that corresponds to a natural language expression, linking linguistic references to fine-grained visual content.

**Region-wise OCR (OCR):** Recognizing and transcribing text within a region, which specified by coordinates or description, facilitating fine-grained interleaved image-text understanding.

**Spatial Relationship Comprehension (SRC):** Understanding geometric relationships (*e.g.*, "above" and "to the left front to") between objects within diverse 3D views, supporting visual-spatial reasoning. Compared to some rudimentary or synthetic spatial understanding datasets (Johnson et al., 2017; Li et al., 2023; Liu et al., 2023b), our data is more realistic in emphasizing spatial location understanding under real-world scenarios as well as 2D images acquired by cameras or cell phones.

**Scene Knowledge Inference (SKI):** Inferring high-level semantic and functional information about the scene or making decision based on the visual contents, incorporating context, commonsense knowledge, and visual cues beyond explicit visual entities. Compared to the regular visual reasoning dataset, `Lens` additionally distinguish between "thought paths" and "final answers", differentiated by the $<think>$ token, aiming to provide finer-grained information for potential test-time scaling tests and R1-style reinforcement learning.

## A.3 QUALITY CONTROL PROCESS

In addition to the annotations, diversified measures were taken to enrich the content of data samples and ensure their quality. Specifically, we implemented a multi-faceted quality control process. Beyond a two-step data cleaning protocol, we also enriched each image with supplementary metadata to facilitate traceability and contextual analysis.

First, we manually performed a two-stage data cleaning process. In the initial stage, we reviewed and eliminated suspected duplicate images. The second stage involved distributing the problems

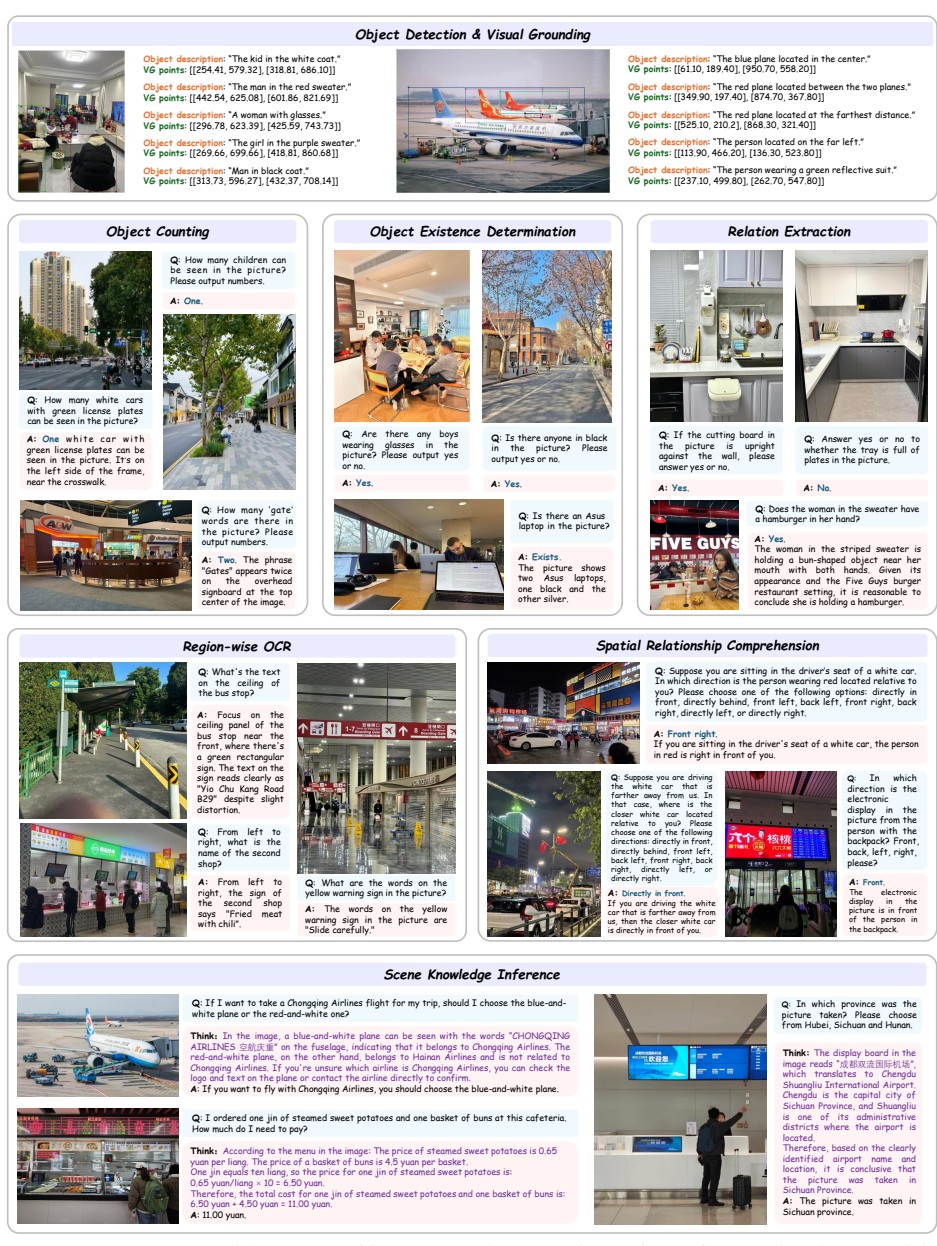

Figure 7: `Lens` covers a wide range of images and annotations, from fine-grained recognition and spatial localization to complex reasoning over extended thought processes. Notably, each image is annotated with labels corresponding to all subtasks concurrently, enabling comprehensive evaluation.

among co-authors for meticulous format and typo checking, ensuring all annotations adhered to a standardized format.

To further validate the quality and consistency of our annotations, we performed an additional two-step verification process. This included both manual and machine-assisted checks. The entire dataset was cross-verified by both an independent team of annotators and the Qwen2.5-VL 72B open-source model. For machine validation, we input the original image, question, and answer into the MLLM. Cases flagged as invalid by the model were isolated for manual re-evaluation by a separate team of annotators. For object detection and visual grounding tasks, we directly visualized the annotations on the images, enabling human evaluators to assess the validity of the bounding boxes.

Additionally, we enriched each image with supplementary metadata. We included a pseudonymized Annotator ID to allow for annotator-specific quality tracking while preserving privacy. The Time of Online Publication and Scene Category were also labeled to facilitate temporal studies, filter outdated content, and organize the dataset by scene. Finally, ambiguous images that consistently resulted in

low inter-annotator agreement were manually filtered out to ensure a high-quality final dataset. Some cleaned representative examples are visualized in Figure 7.

## A.4 DATA PRIVACY PROTECTION AND COPYRIGHT STATEMENT

Our protocol for handling potentially sensitive information was a multi-stage process designed to be as thorough as possible:

**Automated Pre-screening:** As an initial step, we used automated tools (*e.g.*, face detection models , docTR ) to perform a preliminary scan of the collected images. This scan was configured to flag images with a high probability of containing human faces or dense blocks of text that might constitute personally identifiable information.

**Comprehensive Manual Review:** Every image, including those not flagged by the automated scan, was then subjected to a thorough manual review by our team of over 20 trained human annotators. Annotators received specific training and a detailed guide on identifying a wide range of sensitive data, including but not limited to: Visible and recognizable faces; Full names, usernames, or contact information; License plates, street addresses, or other specific location markers; Private documents or screens displaying personal data.

**Sensitive Information Exclusion:** Images containing sensitive personal information were either excluded or processed to blur or mask sensitive regions, to mitigate privacy risks. Based on the manual review, if an image contained sensitive information, one of two actions was taken as mentioned in the paper:

- Processing: If the sensitive information was incidental to the image's main content, we applied irreversible blurring or masking to the specific region.

- Exclusion: If the sensitive information was central to the image and could not be adequately anonymized without destroying the scene's context, the image was entirely excluded from the final dataset.

**Final Verification:** To ensure consistency and quality, a final audit was conducted by a subset of the paper's authors. This team reviewed a random sample of the approved images and 100% of the processed (blurred/masked) images to verify that our privacy protocol was correctly and consistently applied.

This multi-stage, human-centric approach ensures that the images in the `Lens` dataset comply with platform policies and respect individual privacy. The `Lens` dataset is constructed solely for non-commercial academic research purposes and does not transfer copyright ownership of the original images. By accessing or using the `Lens` dataset, users agree to comply with all applicable copyright laws, platform policies, and research-use restrictions.

## A.5 LOW-LEVEL FEATURE ANALYSIS OF IMAGES FROM DIFFERENT SCENES

We counted five low-level visual attributes, including lighting, contrast, color, blur, and spatial information (SI), to assess the statistical difference between different scenes. As shown in Figure 8, the normalized probability density curves of low-level visual attributes across different scenes are consistent with human perceptual preferences. Scenes with regulated lighting conditions (*e.g.*, classrooms, airports, and stations) demonstrate sharp peaks near x $\approx$ 0 in the illumination curves (density $> 0.5$), indicating constrained variations in brightness. In contrast, domestic environments (*e.g.*, living rooms, bedrooms, and kitchens) display broader illumination distributions, suggesting more diverse and adaptive light sources. Furthermore, functional scenes such as bedrooms, bathrooms, and kitchens exhibit sharp, concentrated peaks in color distributions (peak density $\approx 0.5$), implying greater structural regularity in specific visual attributes.

---

https://github.com/timesler/facenet-pytorch
https://github.com/mindee/doctr

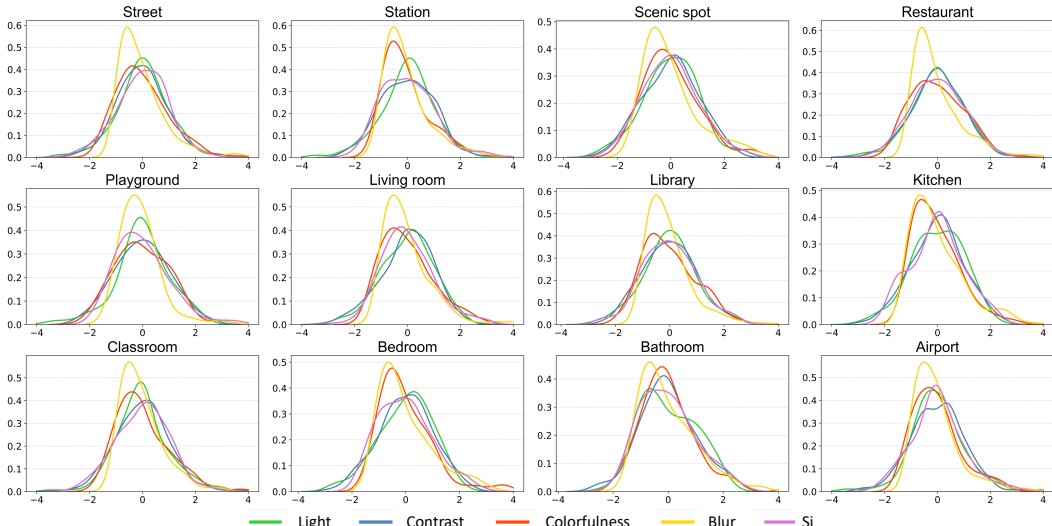

Figure 8: The normalized probability distributions of low-level attributes from different scenes. Scenes with flat peaks show more diversity, while those with sharp peaks have similar features.

## A.6 FINE-GRAINED EVALUATION OF VG

We conducted a fine-grained evaluation of a diverse set of models on the Visual Grounding (VG) task You et al. (2024); Bai et al. (2025); Yao et al. (2025); Zhu et al. (2025); Xiao et al. (2023), categorizing them into two groups: traditional predictive multimodal models and generative multimodal large language models (MLLMs). The results, summarized in Table 4, reveal several key insights into the current state of visual grounding capabilities.

### A.6.1 COMPARISON OF MODEL CATEGORIES

The results clearly indicate a significant performance gap between the two model categories. The top-performing MLLMs, specifically Qwen2.5-VL-7B and Qwen2.5-VL-32B, demonstrate superior performance across all metrics, with an accuracy of 46.94% and 48.47% at $IoU@0.5$, respectively. This performance is substantially higher than the top predictive model, G-DINO, which achieves 37.05% at the same metric. This finding suggests that the generative and in-context learning capabilities of modern MLLMs provide a substantial advantage in the complex VG task, enabling them to better understand nuanced linguistic instructions and ground them accurately in the visual space.

### A.6.2 PERFORMANCE ON DIFFERENT SCALES

A multi-scale analysis, measured by accuracy on small ($ACC_s$), medium ($ACC_m$), and large ($ACC_l$) objects, provides a more granular view of each model's strengths and weaknesses. Both traditional and generative models exhibit a similar trend: performance consistently improves with the size of the target object. This is a common challenge in visual grounding and object detection, as localizing and grounding small objects remains difficult.

**Traditional Models:** Among the traditional models, G-DINO demonstrates a more balanced performance across scales, achieving 24.87% on small objects and 52.32% on large objects. In contrast, models like VLTVG and SimVG struggle significantly with small objects, with accuracies of 0.00% and 0.01% respectively, but show strong performance on large objects (29.70% and 45.20%).

**Generative MLLMs:** While MLLMs also struggle with small objects, their performance is notably better than most traditional models. Qwen2.5-VL-32B and Qwen2.5-VL-7B achieve high accuracy on medium and large objects, with their $ACC_m$ and $ACC_l$ scores reaching 55.04% and 61.48% (for Qwen2.5-VL-32B), and 54.12% and 60.24% (for Qwen2.5-VL-7B) respectively. The strong performance on larger objects may be attributed to their powerful visual backbones and advanced language understanding capabilities, which help them better contextualize the target within the scene.

### A.6.3 IMPACT OF MODEL ARCHITECTURE AND SIZE

Our results also highlight the importance of model architecture and size. The Qwen2.5-VL family of models, with its impressive performance, benefits from a powerful visual encoder (FE-ViT) and a sophisticated Qwen2.5 language backbone. Similarly, the InternVL3 series shows a clear scaling effect, where performance on most metrics improves as the model size increases from 2B to 14B. The performance of the 38B variant is slightly lower than the 14B variant due to its different visual backbone. This trend, consistent with findings in large language models, suggests that scaling up both visual and linguistic components is a promising direction for future research in visual grounding.

| Method | Visual Backbone | Linguistic Backbone | Accuracy @ IoU | | | | | Scale-wise Accuracy | | |
|---|---|---|---|---|---|---|---|---|---|---|
| | | | @0.5 | @0.6 | @0.7 | @0.8 | @0.9 | $ACC_s$ | $ACC_m$ | $ACC_l$ |
| **Methods based on predictive multimodal models:** | | | | | | | | | | |
| TransVG (Deng et al., 2021) | RN101 | BERT-B | 8.73 | 7.57 | 6.29 | 4.40 | 1.69 | 0.01 | 2.01 | 23.64 |
| VLTVG (Yang et al., 2022) | RN101 | BERT-B | 11.04 | 9.60 | 7.75 | 5.33 | 1.99 | 0.00 | 2.80 | 29.70 |
| MMCA (Yao et al., 2024) | RN101 | BERT-B | 10.92 | 9.45 | 7.90 | 5.64 | 2.22 | 0.03 | 2.79 | 29.31 |
| CLIP-VG (Xiao et al., 2023) | CLIP-B | CLIP-B | 8.73 | 7.57 | 6.29 | 4.40 | 1.69 | 0.01 | 2.01 | 23.64 |
| EEVG (Chen et al., 2024c) | ViT-B/16 | BERT-B | 9.27 | 5.78 | 2.51 | 0.48 | 0.05 | 0.01 | 0.98 | 26.12 |
| SimVG (Dai et al., 2024) | BEIT-3 | BEIT-3 | 16.46 | 13.90 | 11.12 | 7.44 | 2.70 | 0.01 | 3.10 | 45.20 |
| G-DINO (Liu et al., 2024b) | Swin-L | BERT-B | 37.05 | 33.92 | 29.20 | 22.57 | 11.36 | 24.87 | 37.54 | 52.32 |
| **Methods based on generative multimodal large language models:** | | | | | | | | | | |
| Groma-7B (Ma et al., 2024) | DINOv2-L | Vicuna | 33.59 | 29.95 | 25.47 | 18.73 | 8.52 | 11.58 | 33.91 | 58.59 |
| Mova-7B (Zong et al., 2024) | Multi-expert | Vicuna | 20.44 | 13.10 | 5.98 | 1.09 | 0.13 | 5.06 | 15.97 | 40.36 |
| Ferret-7B (You et al., 2024) | CLIP-L | Vicuna | 23.26 | 18.97 | 13.95 | 7.61 | 1.84 | 1.85 | 19.49 | 54.64 |
| Ferret-13B (You et al., 2024) | CLIP-L | Vicuna | 24.20 | 19.81 | 14.41 | 8.12 | 2.05 | 2.26 | 20.42 | 56.31 |
| InternVL3-2B (Zhu et al., 2025) | InternViT-0.3B | Qwen2.5 | 7.89 | 5.10 | 2.85 | 1.36 | 0.33 | 0.61 | 3.46 | 19.34 |
| InternVL3-8B (Zhu et al., 2025) | InternViT-0.3B | Qwen2.5 | 17.54 | 13.23 | 8.89 | 4.94 | 1.60 | 3.23 | 15.36 | 35.21 |
| InternVL3-14B (Zhu et al., 2025) | InternViT-0.3B | Qwen2.5 | 29.53 | 23.98 | 17.25 | 10.05 | 3.00 | 4.58 | 27.80 | 57.07 |
| InternVL3-38B (Zhu et al., 2025) | InternViT-6B | Qwen2.5 | 27.85 | 21.42 | 15.00 | 8.23 | 2.37 | 4.91 | 25.81 | 53.56 |
| VLM-R1-3B (Shen et al., 2025a) | FE-ViT | Qwen2.5 | 23.79 | 19.91 | 15.65 | 10.53 | 4.31 | 8.15 | 22.84 | 40.94 |
| Qwen2.5-VL-3B (Bai et al., 2025) | FE-ViT | Qwen2.5 | 45.03 | 37.92 | 29.33 | 18.48 | 6.51 | 29.14 | 50.54 | 57.15 |
| Qwen2.5-VL-7B (Bai et al., 2025) | FE-ViT | Qwen2.5 | 46.94 | 39.39 | 29.94 | 18.38 | 6.26 | 28.87 | 54.12 | 60.24 |
| Qwen2.5-VL-32B (Bai et al., 2025) | FE-ViT | Qwen2.5 | 48.47 | 40.66 | 30.78 | 19.15 | 6.63 | 30.93 | 55.04 | 61.48 |

Table 4: Multi-scale evaluation results.

### A.7 MORE SYNERGISTIC EFFECTS ANALYSIS

We further notice that Figure 6 (a) shows OC-OCR correlation (0.77) ≫ OC-OE (0.46). This contradicts intuition—object counting (OC) should align more naturally with existence checks (OE) than OCR. We attribute this to two factors. First, the Object Existence (OE) task is a simple binary classification: either an object is present or it is not. It requires a model to make a broad, scene-level assessment. In contrast, Object Counting (OC) is a more demanding task that requires fine-grained localization of individual objects, followed by an enumeration step. A model can be highly proficient at a binary existence check without possessing the precise localization and counting skills needed for the OC task. This fundamental difference in cognitive demand limits the correlation between the two. Second, the high correlation between Object Counting (OC) and OCR is not coincidental. Both tasks rely on a critical shared capability: fine-grained localization. To perform well on the OC task, a model must accurately identify and localize each instance of an object to count it. Similarly, to perform region-wise OCR, the model must first precisely locate the bounding box of the text before reading it. The strong correlation suggests that the ability to perform precise object localization is a dominant factor in a model's success on both tasks, thus strengthening their relationship despite their different end goals.

To further test the synergy between different tasks, we conducted a experiment with Qwen2.5-VL-7B on a sampled subset of `Lens`. Specifically, when testing the SKI task, we fed the VQA question-answer pairs of other tasks into the model as context along with the question, and asked it to return the answer (refer to Appendix A.14 for the prompt template $p_s$, where we provide an example based on the OCR task). The test results are shown in Table 5 and reveal several noteworthy patterns. First, incorporating OCR information yields a substantial performance gain (from 39.80% to 41.36%), indicating that understanding scene text helps the model solve some reasoning tasks. Second, although some tasks—such as OE and OC—exhibit limited or even negative effects when introduced individually, their combination with OCR consistently boosts performance. This may indicate that auxiliary perceptual signals, while insufficient on their own, can enhance reasoning when

mediated through textual understanding. Furthermore, the observed synergistic effects resonate with the design philosophy of our proposed Self-Driven Multi-Expert Collaborative (SMEC) framework. We argue that complex multimodal reasoning cannot be achieved by isolated competencies alone, but requires a structured mechanism to coordinate heterogeneous sources of evidence.

| OCR | - | ✔ | - | - | - | ✔ | ✔ | ✔ | ✔ | ✔ | ✔ |
| RE | - | - | ✔ | - | - | ✔ | - | - | ✔ | ✔ | ✔ |
| OE | - | - | - | ✔ | - | - | ✔ | - | ✔ | - | ✔ |
| OC | - | - | - | - | ✔ | - | - | ✔ | - | ✔ | ✔ |
| Performance | 39.80 | 41.36 | 38.03 | 39.23 | 39.52 | 40.72 | 41.45 | 40.73 | 41.93 | 40.79 | 41.90 |

Table 5: Testing the synergistic effects of different tasks on Scene Knowledge Inference (SKI).

## A.8 ANALYSIS OF INPUT RESOLUTION

| Method | Visual Backbone | Linguistic Backbone | Accuracy @ Input Resolution | | | |
| --- | --- | --- | --- | --- | --- | --- |
| | | | $640 \times 640$ | $960 \times 960$ | $1280 \times 1280$ | $1600 \times 1600$ |
| InternVL3-2B (Zhu et al., 2025) | InternViT-0.3B | Qwen2.5 | 40.97 | 41.53 | 41.90 | 40.60 |
| InternVL3-9B (Zhu et al., 2025) | InternViT-0.3B | InternLM3-8B | 46.95 | 46.54 | 46.65 | 46.63 |
| InternVL3-14B (Zhu et al., 2025) | InternViT-0.3B | Qwen2.5 | 50.28 | 51.15 | 51.58 | 51.17 |
| InternVL3-38B (Zhu et al., 2025) | InternViT-6B | Qwen2.5 | 50.88 | 50.97 | 49.98 | 51.08 |
| Qwen2.5-VL-3B (Bai et al., 2025) | FE-ViT | Qwen2.5 | 40.08 | 40.44 | 40.48 | 40.58 |
| Qwen2.5-VL-7B (Bai et al., 2025) | FE-ViT | Qwen2.5 | 46.52 | 47.41 | 48.13 | 48.11 |
| Qwen2.5-VL-32B (Bai et al., 2025) | FE-ViT | Qwen2.5 | 53.72 | 54.37 | 54.10 | 54.09 |
| GLM-4.1V-Base-9B (Hong et al., 2025) | AlMv2-Huge | GLM-4-0414 | 42.35 | 42.86 | 43.28 | 43.73 |
| GLM-4.1V-Thinking-9B (Hong et al., 2025) | AlMv2-Huge | GLM-4-0414 | 48.77 | 50.78 | 51.32 | 51.13 |

Table 6: Benchmark results across varying input resolutions.

We conducted a detailed analysis to understand the impact of varying input resolutions on model performance. The results, summarized in Table 6, reveal several key insights.

**General Trend (Performance Improves with Resolution):** For most models, performance generally improves as the input resolution increases. This trend is evident in models such as Qwen2.5-VL-7B, which shows a steady increase in accuracy from 46.52% at 640×640 to 48.13% at 1280x1280. Similarly, GLM-4.1V-Thinking-9B improves from 48.77% to 51.32% over the same range. This is expected, as higher resolutions provide more visual detail, which is particularly beneficial for complex visual grounding and reasoning tasks that require fine-grained perception.

**The Point of Diminishing Returns:** However, the results also suggest a point of diminishing returns. For many models, the performance gain from increasing the resolution beyond 1280x1280 is minimal, and in some cases, performance slightly decreases. For example, InternVL3-14B peaks at 51.58% at 1280x1280 and then slightly drops to 51.17% at 1600x1600. Similarly, Qwen2.5-VL-7B's performance plateaus at 1280×1280. This phenomenon could be attributed to several factors, including the model's architecture, which may not be fully optimized to handle the extra high-resolution information, or the fact that the added detail does not contribute meaningfully to solving the task.

**Model-Specific Variations:** Interestingly, some models, like InternVL3-2B, show less sensitivity to resolution changes, with its performance remaining relatively stable across all resolutions. In contrast, models such as GLM-4.1V-Thinking-9B and Qwen2.5-VL-32B demonstrate a more pronounced performance improvement with higher resolutions, indicating that their architectures are more capable of leveraging the extra visual information. This suggests that the optimal input resolution is not a one-size-fits-all solution and depends heavily on the model's architecture and design.

## A.9 QUALITATIVE ERROR ANALYSIS

To better illustrate common failure patterns and the underlying limitations of current Multimodal Large Language Models, we conduct a qualitative analysis of representative error cases across different task levels. Following the structure of our benchmark, we group the visualizations into two categories: (1) VQA-style tasks and (2) Localization tasks, including detection and visual grounding, as shown in Figure 9.

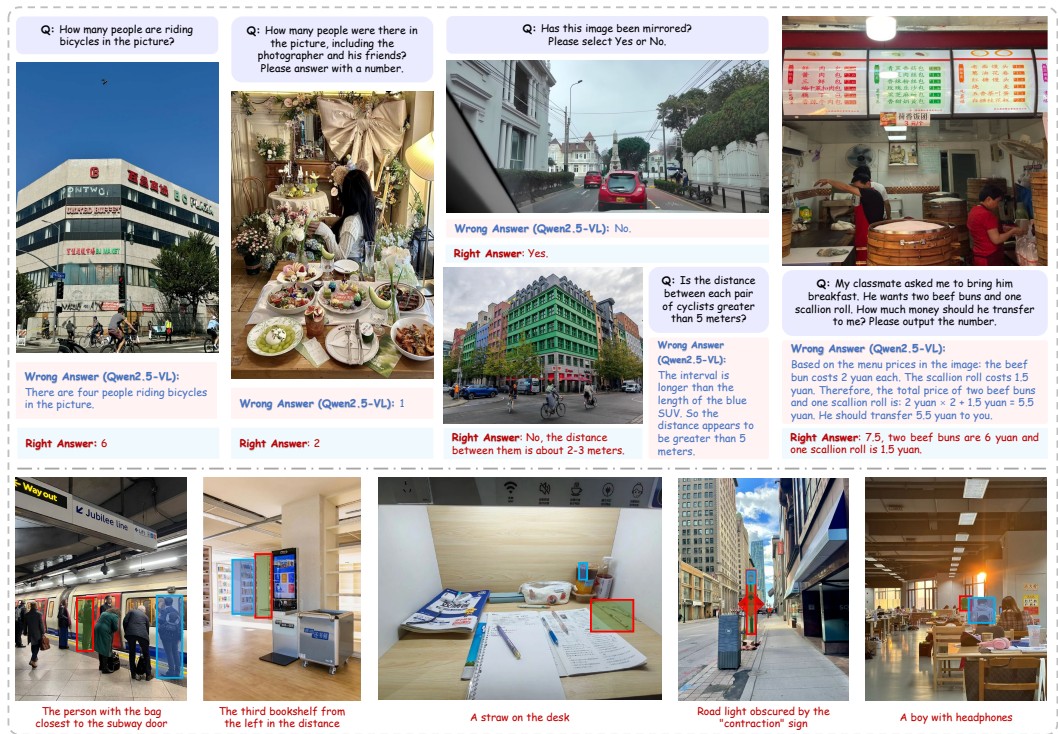

Figure 9: Failure cases for various tasks. Incorrect predictions and labels are indicated by blue and red, respectively

For Perception & Understanding Tasks, they primarily require directly aligning visual content with textual queries. While modern MLLMs achieve reasonably high accuracy, their errors frequently stem from low-level perceptual limitations such as small objects, occlusion, distant subjects, and sensitivity to resolution. Several examples clearly illustrate this issue—for instance, miscounting the number of cyclists in a street scene, or failing to detect objects like a straw on a desk or a road light partially obscured by signage. For more complex reasoning tasks, we observe two major classes of systematic errors.

**Correct reasoning, incorrect perception:** The model often demonstrates sound logical reasoning but bases its inference on incorrect or incomplete visual extraction. For example, in a price-computation task, although the model performs the arithmetic correctly, it misreads the menu price of the beef bun, leading to a wrong total (5.5 instead of 7.5 yuan). This reveals a persistent bottleneck where high-level reasoning is constrained by low-level perception, especially OCR and fine-grained attribute recognition. Such cases motivate the need for agent-based or expert-collaborative pipelines, which can iteratively refine visual cues or invoke specialized perception experts. They also highlight the importance of dynamic zoom-in strategies for capturing critical but small textual or visual details. (Wu & Xie, 2024; Zhang et al., 2025a; Shen et al., 2025b; Shao et al., 2024)

**Spatial and physical reasoning deficits:** A second recurring failure mode involves questions requiring geometric or physical commonsense. Models frequently struggle with tasks that implicitly require depth understanding, object-scale priors, or spatial metric reasoning. For instance, the model incorrectly concludes that two cyclists are more than 5 meters apart by comparing them to the length of a blue SUV, even though the correct distance is only about 2–3 meters. Likewise, it fails to judge whether an image is mirrored due to misunderstanding spatial layout cues. These issues echo recent findings showing that current MLLMs still lack robust spatial grounding and physical commonsense. (Zhang et al., 2025b; Azzolini et al., 2025)

## A.10    A FORMAL DESCRIPTION OF SMEC

As shown in Algorithm 1. A key advantage of SMEC is that it does not depend on fixed, hand-crafted prompts. Instead, the prompts and expert descriptions are self-generated by the model based on

the given visual input and question. During each iteration, the model adaptively refines its expert descriptions and updates the generation process when redundancy or low-quality information is detected. This adaptive design means that SMEC is not tied to a specific phrasing or a predefined set of experts, but can flexibly adjust to different problems and question types. As a result, our method is more robust than approaches that rely heavily on manually designed prompts, since the "experts" in SMEC emerge dynamically from the model itself rather than being externally imposed.

---

**Algorithm 1** Self-driven Multi-Expert Generation & Collaboration

---

**Initialization**: Based Instruction-tuned MLLM $\theta$, Question $q$, Meta Generation Prompt $p_g$, Inspection prompt $p_i$, Collaboration Prompt $p_c$, Description Set $D = \emptyset$, Maximum Answer Set $A = \emptyset$, Iterations $N_t$.

1:    $a_0 = \theta(q), A = A \cup a_0$            # Initial answer for question.
2:    **for** $t = 1, 2, \ldots, N_t$ **do**
3:      **if** $t = 1$ **then**
4:        $d_q^1 = \theta(p_g, q, A_0)$            # Initial expert description.
5:      **else**
6:        $d_q^t = \theta(p_g, D, q, A_t)$     # New description based on existing information.
7:      **end if**
8:      **if** $\theta(p_i, D, d_q^t) = Retain$ **then**
9:        $D = D \cup d_q^t$            # Checking process.
10:       $a_t = \theta(q, d_q^t), A = A \cup a_t$     # New answer from the expert perspective.
11:      **else**
12:       $p_g = \theta(q, d_q^t, p_g, D)$ # Update generation prompt while repeat descriptions.
13:      **end if**
14:    **end for**
15:    $a_{final} = \theta(q, A, p_c, D)$            # Summarize the final answer.

---

## A.11   HUMAN PREFERENCE

To ensure the verifiability of our evaluation, particularly for open-ended reasoning tasks, we employed a large language model (LLM) as an automatic grader. To mitigate the concern regarding potential LLM hallucinations or failure to detect nuanced mistakes, it is noted that the LLM grader (*e.g.*, GLM4-flash) is used to compare the model-generated responses against our pre-existing, human-annotated answers, ensuring that the ground truth remains anchored in high-quality human data. We also conducted a human preference analysis on a representative and complex task SKI with a subset of the dataset, aiming to provide a gold standard against which to measure the reliability of judgement model. We recruited a separate team of ten expert annotators who were not involved in the original data collection to manually evaluate the accuracy of the model's answers compared to the labeled answers, as shown in Table 7. In this setup, Actual Positive (AP) and Actual Negative (AN) represent human judgments of correctness and incorrectness, respectively, while Test Positive and Test Negative indicate the LLM grader's corresponding evaluations.

The results, shown in Table 7, demonstrate a strong alignment between human preference and LLM-based judgments across all evaluated models. For instance, in the case of Qwen2.5-VL's responses, when humans labeled an answer as correct (AP), the LLM grader also marked it as correct 97.13% of the time. Similarly, when humans judged an answer as incorrect (AN), the LLM grader agreed 96.14% of the time. Comparable trends are observed for InternVL3 and Gemini2.5-pro, though with slightly larger gaps in negative cases. These findings suggest that the LLM grader provides a highly reliable approximation of human judgment, especially for positive cases. Incorporating human validation thus not only confirms the feasibility of using LLMs as evaluators but also highlights their potential to scale evaluation consistently across large datasets while retaining strong alignment with expert human preference.

## A.12   TEMPORAL GENERALIZATION PERFORMANCE

We visualize the accuracy of the best few models on data from different time periods in Figure 10, including Qwen2.5-VL (7B/72B), InternVL3 (9B/78B), and Gemini-2.5-Pro. Notably, models released after late 2024, such as InternVL3-78B and Qwen2.5-VL-72B, consistently outperform

| Method | Qwen2.5-VL | | InternVL3 | | Gemini2.5-pro | |
| --- | --- | --- | --- | --- | --- | --- |
| | AP | AN | AP | AN | AP | AN |
| **Test Positive** | 97.13% | 2.87% | 94.70% | 5.30% | 93.65% | 6.35% |
| **Test Negative** | 3.86% | 96.14% | 7.77% | 92.23% | 11.41% | 88.59% |

Table 7: Human evaluation for the models' responses.

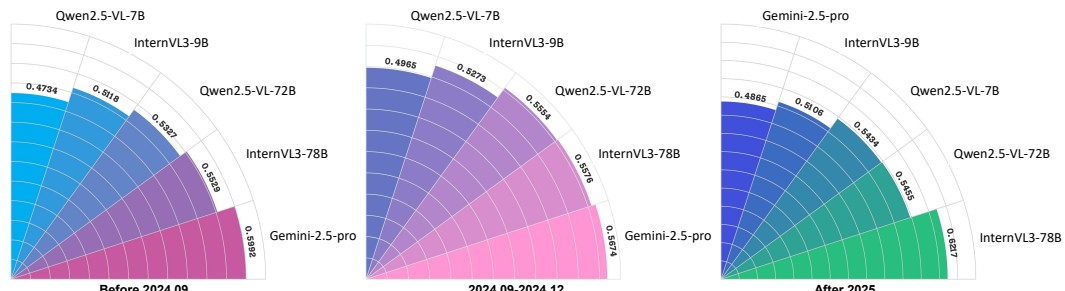

Figure 10: Model accuracy on SKI task across temporal split.

Gemini-2.5-pro on contemporary data, although their overall performance lower than Gemini-2.5-pro. The experimental results further supporting the observation that newer models tend to generalize better to new data distributions. This trend underscores the impact of scaling, instruction tuning, and exposure to temporally aligned data in enhancing multimodal reasoning performance.

## A.13 More on SMEC vs. Baseline methods

To further highlight the advantage of our proposed SMEC framework, we provide additional qualitative comparisons against baseline methods, including direct inference and ChatGPT-style single-pass reasoning. Figure 11 showcases representative challenging samples drawn from `Lens`, where baseline models tend to produce either incomplete or overconfident predictions.

In these cases, ChatGPT and other baselines often failed for two recurring reasons:

**Over-Reliance on Surface Cues.** Baselines typically produced answers anchored on the most salient visual elements, neglecting contextual or relational signals. For instance, when asked to infer spatial constraints or traffic rules in a airport photo, ChatGPT tended to extrapolate directly from textual OCR cues, leading to plausible but incorrect answers.

**Lack of Internal Deliberation.** Without multi-perspective reasoning, baselines converged prematurely on a single hypothesis. This often caused brittle errors in scenarios requiring integration of textual, spatial, and commonsense evidence.

By contrast, SMEC decomposed the problem into multiple role-specific perspectives, such as a scene analyst, a spatial reasoner, and a cultural or commonsense expert. These experts generated partially overlapping but complementary hypotheses, which were then screened for redundancy and synthesized into a consensus. In the traffic-sign example shown in Figure 11, SMEC correctly filtered out spurious cues and converged on the right driving instruction, whereas ChatGPT remained uncertain or hallucinated unsupported details. Across difficult samples, two consistent patterns emerged:

**Error Correction through Redundancy Filtering.** Even when some experts produced misleading interpretations, SMEC's screening stage effectively down-weighted unreliable reasoning chains, preventing error propagation.

**Multi-Step Enrichment.** Iterative expert collaboration yielded richer reasoning trajectories, allowing the framework to exploit synergies between visual grounding, OCR, and commonsense inference. This process systematically improved robustness to ambiguous or noisy inputs.

Taken together, these results suggest that SMEC is not merely a test-time ensemble but a principled framework that encourages internal debate within a single model. Unlike single-pass inference, SMEC operationalizes a language-native form of deliberation, enabling models to approximate the

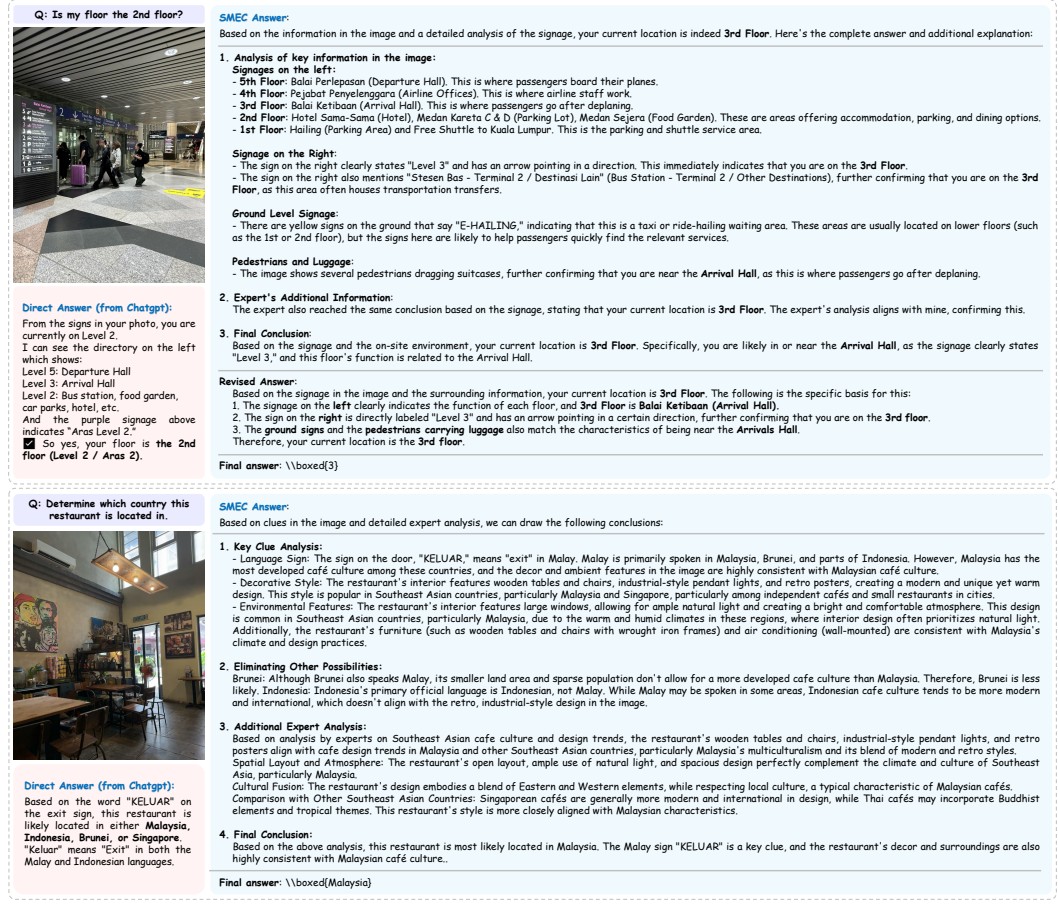

Figure 11: Comparison of our method and ChatGPT on some difficult examples.

dynamics of human expert panels. As Figure 11 illustrates, this mechanism directly translates into more accurate and interpretable reasoning on complex multimodal questions.

## A.14 Prompts for Synergy Testing

**Prompt $p_s$ with OCR question-answer pairs as context**

$\{question\ from\ SKI\}$
You can use the following facts to help you answer this question. Please note that they may not be relevant to the question. Here are some factual questions and answers about this picture:
$\{question\ from\ OCR\}$: $\{answer\ from\ OCR\}$

### A.15 PROMPTS OF SELF-DRIVEN MULTI-EXPERT COLLABORATIVE FRAMEWORK

---

**Meta Generation Prompt $p_g$**

$\{question\}$
Here are your answers and those of some experts:
$\{answer\}$
Now you can create and work with multiple experts to improve your answer. So, please describe in as much detail as possible the different skills and focus you need from each expert. We will provide each expert with the same information and queries. Each expert should have his or her own specialization covering perception, understanding and reasoning, etc., so you can assign only one subtask to each expert to ensure a more refined answer. We will relay their responses to you in turn so that you can reorganize them into better answers. Please note that descriptions should be in the second person, e.g. You are XXX.
These are the descriptions of the experts you have previously created for this task:
$\{description\}$
Therefore, do not create the same experts as above over and over again.
Now you can create a description for the new expert (please note that you can only describe one, not more than one at the same time):

---

**Inspection Prompt $p_c$**

$\{question\}$
We hired multiple experts to answer this question. Below is a second person description of the experts we hired: $\{existing\ description\}$
We are now hiring a new expert to help better provide the information needed for the question as well as respond to user queries. Here is a second person description of the new expert: $\{description\}$
Since there is an additional cost to hiring a new Expert, please evaluate the new Expert based on the following two criteria to decide whether or not to retain them.
1. based on the new Expert's description, determine if they can effectively assist in answering the user's question or provide the information needed for the question.
2. the new expert is not a duplicate of any existing expert.
The new expert must meet both of these criteria. If either criterion is not met, they should be discarded. If retaining, please reply 'Retain'. If discarded, please reply: 'Discard'.

---

**Collaboration Prompt $p_c$**

$\{question\}$ These are you and some experts' answer: $\{answer\}$
The description of the experts you invited are: $\{description\}$
Now, you can refine your answer based on the answer and additional information they provided to better answer the question. Keep in mind that the experts' answer and additional information may not be correct, so decide carefully whether to accept his answer or stick to your original one.
Revised answer:

---

### A.16 LIMITATIONS

While `Lens` offers broad task coverage and a unified evaluation setting, it currently focuses on static images and short-form reasoning. Real-world applications may require multimodal reasoning over temporal sequences or long-form narratives, which are beyond the scope of this version. Additionally, SMEC relies on prompt-based expert simulation, which, though flexible, may introduce redundancy or sensitivity to prompt phrasing and inference.

### A.17 THE USE OF LARGE LANGUAGE MODELS (LLMS)

We have not used Large Language Models (LLMs) for our paper writing.

