# OpenReview forum: "LENS: Multi-level Evaluation of Multimodal Reasoning with Large Language Models"
_ICLR.cc/2026/Conference — ICLR 2026 Poster_

### Official Review · Reviewer_4Zew · 2025-10-16

**Soundness:** 2
**Presentation:** 1
**Contribution:** 1
**Rating:** 2
**Confidence:** 2

**Summary:**

This paper introduces Lens, a benchmark for evaluating MLLMs on multi-modal reasoning and SMEC, a framework for MLLMs to solve the tasks with self-generated expert opinions. The Lens consists of eight tasks under three main categories: perception, understanding and reasoning. Images (3.4K) of the benchmark are collected manually from variety social media sources. The results of the experiments on Lens show that low-level visual tasks can affect the higher-order reasoning process. When the models apply SMEC for Scene Knowledge Inference task, the accuracy of the models increases with more iterations.

**Strengths:**

•	The data is manually filtered and annotated by human verifiers.

•	The process of SMEC is explained very detailly.

•	Synergistic effects evaluation provides detailed examination of cross-tasks performance.

**Weaknesses:**

•	The overall writing quality of the paper could be improved. Some sections lack clarity, and certain sentences are difficult to understand.

•	While the paper highlights multi-level evaluation and integrated tasks as key contributions, these concepts are not sufficiently explained or exemplified in the paper.

•	The paper does not clearly justify the need for introducing a new benchmark. Several existing benchmarks already include some of the tasks used in the paper. It remains unclear why extending existing benchmarks with additional tasks would not be sufficient to analyze cross-task interactions.

•	The authors did not explain why the community needs a new benchmark. Other benchmarks have already included some tasks. By adding other task-related questions to the existing benchmarks and data, the same effect can be achieved. In that case, the evaluation of the synergistic effect does not require generating a new benchmark.

•	It remains unclear how the proposed SMEC framework distinguishes itself from prior work in terms of novelty and contribution. The related work section does not address prior efforts involving expert-based or modular task-solving frameworks.

•	The experimental validation of SMEC appears limited, as its performance is only evaluated on a single task (Scene Knowledge Inference). To validate its effectiveness, it would be valuable to test SMEC across other tasks and other datasets.

**Questions:**

•	In which task is the interleaved image-text feature used? Figure 7 does not show the example.

•	The second link on foot page opens Instagram webpage, not developer agreement. The reviewer didn’t understand the relation of the link and text in the paper.

•	Do the authors get legal permissions from social media platforms X, Instagram, Weibo and RedNote to use their data? Although some processes, like erasing facial information, have been done for privacy concerns, the images are still copyrighted. For example, the copyright regulations of RedNote state that “RedNote's trademarks, logos, and content are protected by intellectual property laws. You may not use our intellectual property without explicit permission.” According to these cases the authors should get permission from the platforms to use their data for research purposes. If permission has been granted for data use, including documentation of this in the Appendix, is recommended.
RedNote: https://red-note.co/terms-of-service

•	It would be valuable to report human performance on the Lens benchmark and the performance comparison between humans and models.

•	The Related Work section does not clearly articulate how Lens differs from existing benchmarks, particularly in terms of visual and reasoning capabilities. It would be helpful to clarify what unique contributions or challenges Lens introduces that are not addressed by prior datasets.

Suggestions:

•	The organization of the paper could be improved. For instance, placing the 'Related Work' section in the Appendix makes it harder to assess the motivation and novelty, whereas some less critical elements like Figure 2 are included in the main text

**Details Of Ethics Concerns:**

The authors collected images from social media platforms, X, Instagram, Weibo, and RedNote. Although they stated in the paper that "During the collection process, we strictly complied with the copyright and licensing regulations of each platform, ensuring that data was collected only from publicly accessible posts and that no images were downloaded from sources explicitly prohibiting data reuse or redistribution.", they only shared X's policy. The copyright regulations of RedNote state that “RedNote's trademarks, logos, and content are protected by intellectual property laws. You may not use our intellectual property without explicit permission.” According to this case, the authors might need to get legal permission from the platform to use their data for research purposes.

RedNote: https://red-note.co/terms-of-service

---

> ### Author Response · Authors · 2025-11-20
> **Response to Reviewer 4Zew (Part I):**
>
> ### **W1. Overall writing quality**
>
> **A1.** Could you please specify which sentences or parts are difficult to understand? We would be happy to clarify or revise them accordingly.
>
> ---
>
> ### **W2. Insufficient explanation of multi-level evaluation concepts**
>
> **A2.** Unlike prior task-specific datasets, LENS uses a **unified image set**—3.4K high-resolution, social media images—paired with 60K+ human-curated questions across 12 real-world scenarios. Each image is annotated for **eight tasks**, organized into **three progressive tiers**: *perception*, *understanding*, and *reasoning*.
>
> This design enables a structured evaluation of how lower-level perceptual skills contribute to higher-level reasoning, while minimizing distributional inconsistencies commonly seen in task-specific datasets. We argue that this setup further allows detailed examination of **synergistic effects across tasks**.
>
> ---
>
> ### **W3 & W4. Motivation for the LENS benchmark**
>
> **A3/4. They seem to be the same problem. We address these two concerns together.**
>
> As stated in the abstract, existing benchmarks are typically **task-oriented**, without ensuring that different tasks originate from a consistent underlying data distribution. As a result, they provide limited ability to evaluate how lower-level perceptual capabilities influence higher-order reasoning. Moreover, existing classic datasets have already been heavily used as training corpora, making them unsuitable for evaluating synergistic effects or generalization. In contrast:
>
> * Our images are **brand new**, collected in 2024–2025.
> * All annotations are **manually curated**, requiring substantial effort and cost.
> * Each image supports all **eight tasks**, which few datasets achieve.
> * Since LLMs/VLMs take months to train, our images are effectively **unseen** for most MLLMs, reducing the risk of data contamination.
> * LENS additionally distinguish between **thought paths** and **final answers** for SKI task.
>
> Therefore, LENS fills an important gap by providing a clean, contemporary, multi-task, multi-level benchmark with rich annotations:
> | Capability / Property                                                                      | Existing Benchmarks | LENS  |
> | ------------------------------------------------------------------------------------------ | ------------------- | ----- |
> | **Unified multi-task annotation on the same images**                                       | ✗                   | **✓** |
> | **Three-level hierarchical capability structure (perception → understanding → reasoning)** | ✗                   | **✓** |
> | **Cross-task synergy evaluation**                                                          | ✗                   | **✓** |
> | **Complex reasoning grounded in realistic everyday contexts**                              | limited             | **✓** |
> | **Detailed thought process for real-world tasks (SKI)**                                | partial                   | **✓** |
>
> ---
>
> ### **W5 & W6. Motivation and experimental validation of SMEC**
>
> **A5/6.** SMEC uses the **same MLLM**, prompted with **multiple role-specific instructions**, to collaboratively solve complex reasoning tasks. It is specifically designed for the **reasoning tier**, and especially for the **Scene Knowledge Inference (SKI)** task in LENS.
>
> As shown in Fig. 7, SKI questions require the model to integrate user instructions with complex real-world reasoning—e.g., selecting appropriate transportation, computing consumer prices, or inferring the location of a photograph. SMEC aims to enhance performance on these tasks through **test-time scaling** using **self-generated role-playing prompts**.
>
> There is no particular motivation to apply SMEC to lower-level tasks such as spatial understanding, localization, or OCR, which primarily depend on perceptual grounding rather than multi-agent reasoning.
>
> Our main goal is to provide:
>
> 1. **A high-quality dataset and benchmark** for multi-level reasoning, and
> 2. **potential research directions** based on our benchmark.
>
> Evaluating SMEC on other datasets is beyond the scope of this work and not our primary focus.

---

> ### Author Response · Authors · 2025-11-20
> **Response to Reviewer 4Zew (Part II):**
>
> ### **Q1. The interleaved image–text feature in Figure 7**
>
> **A1.** We follow prior definitions of *interleaved image–text data* as described in [1–5]. In our benchmark, both **Region-wise OCR** and **Scene Knowledge Inference** exhibit this property. These tasks require the model to jointly interpret visual content and embedded textual elements—such as identifiers, menus, and brand logos—to determine, for example, the location of a sign, the price of a product, or the airline of an aircraft. Such samples naturally reflect the interleaved image–text characteristic because textual information is visually grounded within the image itself.
>
> **References**
>
> [1] *SEED-Bench: Benchmarking Multimodal Large Language Models*, CVPR 2024
>
> [2] *Obelics: An open web-scale filtered dataset of interleaved image-text documents*, NeurIPS 2023
>
> [3] *Learning Interleaved Image-Text Comprehension in Vision-Language Large Models*, ICLR 2025
>
> [4] *CoMM: A Coherent Interleaved Image-Text Dataset for Multimodal Understanding and Generation*, CVPR 2025
>
> [5] *MM-Interleaved: Interleaved Image-Text Generative Modeling via Multi-modal Feature Synchronizer*, arXiv 2024
>
>
> ---
>
> ### **Q2. Clarification on the second link in the footnote**
>
> **A2.** The second link points to the **official developer documentation for the Instagram Platform**:
>
> * Instagram Platform (overview): [https://developers.facebook.com/products/instagram/](https://developers.facebook.com/products/instagram/)
> * Developer documentation: [https://developers.facebook.com/docs/instagram-platform](https://developers.facebook.com/docs/instagram-platform)
>
> ---
>
> ### **Q3. Ethics concerns**
>
> **A3.** We appreciate the reviewer’s attention to data ethics. Our dataset adheres strictly to non-commercial and privacy-preserving principles:
>
> * We only use the **image content itself** from publicly viewable posts.
> * All annotations are **human-authored**, manually curated, and *not* derived from user captions, comments, or metadata.
> * Data collection is fully **non-automated**, ensuring no scraping and no access to user information.
> * Our goal is to ensure timeliness and realism of real-world scenarios, not to leverage any user-specific information.
> * Following established practice in LAION-400M, Conceptual Captions, and COYO, we can also only release:
>   **(i)** URLs of publicly visible images, and
>   **(ii)** our annotations,
>   **without** distributing raw image files.
>
> **References**
>
> [1] *Laion-400m: Open dataset of clip-filtered 400 million image-text pairs*, arXiv 2021
>
> [2] *Conceptual captions: A cleaned, hypernymed, image alt-text dataset for automatic image captioning*, ACL 2018
>
> [3] *COYO-700M: Image-Text Pair Dataset*, https://github.com/kakaobrain/coyo-dataset
>
> ---
>
> ### **Q4. Human performance on the Lens benchmark**
>
> **A4.** Our benchmark focuses on multi-level perception-to-reasoning tasks rooted in **common real-world knowledge**, such as route planning, finding flights, or computing basic shopping costs. These questions are intentionally designed to be answerable by non-experts. Since the dataset is itself annotated by humans (who provide the ground truth), the expected human accuracy is extremely high.
>
> ---
>
> ### **Q5. Related Work placement**
>
> **A5.** Our unique contributions are primarily discussed in the **Introduction** and summarized in **Table 1**, which position our benchmark relative to prior work. We also discussed it in A3/4 for weakness. We placed the full Related Work section in the appendix to allocate more space in the main paper for core ideas and empirical findings, an approach commonly adopted in ICLR submissions [1-2].
>
> In the revision, we will additionally include a short paragraph in the introduction to better highlight the distinctions from related work.
>
> **References**
>
> [1] *Navigating the Digital World as Humans Do: Universal Visual Grounding for GUI Agents*, ICLR 2025 (Oral)
>
> [2] *Measuring and Enhancing Trustworthiness of LLMs in RAG through Grounded Attributions and Learning to Refuse*, ICLR 2025 (Oral)

---

> ### Comment · Reviewer_4Zew · 2025-11-26
> **Response to Authors**
>
> Thank you for the clarification of some of my questions. However, most of my major concerns remain unaddressed.
>
> - Although one of the contributions of the study is SMEC, a self-driven multi-expert collaborative framework, the related work section does not address prior efforts involving expert-based or modular task-solving frameworks. This makes it difficult to assess the novelty and contribution of the proposed framework.
> - The authors fail to validate SMEC's effectiveness on other complex real-world reasoning datasets, stating that it is beyond the scope of this work and not their primary focus. Consequently, the reliability and practical effectiveness of SMEC remain uncertain. Without quantitative analysis from other datasets, the SMEC's performance contribution is limited by the Scene Knowledge Inference task in LENS and has not been verified by other benchmarks.
> - The paper’s definition of interleaved image–text data does not match the standard usage in the datasets [1–5] authors provided. These datasets provide captions or textual input alongside images. In contrast, the authors process interleaved-image text as simply interpreting textual elements within the image itself (identifying the logo), without any additional text. Since these examples do not align with the established meaning of interleaved image–text data, characterization of these tasks as interleaved appears conceptually inconsistent and weakens the conceptual clarity of the benchmark. Moreover, this concept is stated as a distinct feature of the proposed dataset in Table 1.
> - I thank the authors for their detailed clarification of the dataset's non-commercial and privacy-preserving design. However, the response does not address the copyright and platform policy issue. Even though the collected data is publicly available, the platforms' terms of service might restrict use without explicit permission, like in RedNote. Referencing other datasets for data release does not constitute legal compliance.
> - Human annotation of the dataset does not fulfill the request for human evaluation. The annotation is a distinct process from evaluation. The authors state that "benchmark helps identify the gaps between current model capabilities and the requirements of human-aligned reasoning systems". To validate how questions are aligned with human perception, understanding, and reasoning, it requires human performance evaluation. Claiming that "human accuracy is extremely high because the human annotation process" remains speculative without measured results.
>
> Because these key issues remain unresolved, I will increase my score slightly.

---

> > ### Author Response · Authors · 2025-11-29
> > **Response to Reviewer 4Zew**
> >
> > **First, we sincerely thank the reviewer for being willing to raise the score. We address each of your remaining concerns point by point below:**
> >
> > **R1.** To further illustrate the novelty and contribution of our method, we will add a dedicated paragraph in the rebuttal revision that situates **SMEC** within the broader landscape of **agentic reasoning, multi-expert/agents collaboration, and modular LLM frameworks**. This addition will better contextualize our contributions and clarify the novelty of our self-driven multi-expert collaborative design.
> >
> > ---
> >
> > **R2.** To further address your concerns, we additionally validate SMEC on a real-world multimodal reasoning benchmark, **RealWorldQA** [1], to demonstrate its broader applicability. We selected this dataset because it evaluates **real-world common-sense reasoning** and has been widely adopted by Qwen-VL, InternVL, and other state-of-the-art MLLMs.
> >
> > The results below show that SMEC consistently improves performance across both the 3B and 7B variants of Qwen2.5-VL, indicating that the framework generalizes beyond the LENS dataset and provides clear benefits on real-world reasoning tasks:
> >
> > | Method   | Model           | RealWorldQA$_{avg}$ |
> > | -------- | --------------- | --------------- |
> > | Direct   | Qwen2.5-VL (3B) | 65.4            |
> > | **SMEC** | **Qwen2.5-VL (3B)** | **67.5**       |
> > | Direct   | Qwen2.5-VL (7B) | 68.5            |
> > | **SMEC** | **Qwen2.5-VL (7B)** | **70.4**      |
> >
> > These improvements confirm that SMEC offers meaningful gains beyond the LENS benchmark and remains effective on complex real-world multimodal reasoning tasks. We will incorporate this analysis into the revised manuscript.
> >
> > [1] X.AI. *Grok-1.5 vision preview*. https://x.ai/blog/grok-1.5v, 2024.
> >
> > ---
> >
> > **R3.** We would like to clarify that each image in our benchmark is accompanied by **rich textual information**. As we did in **Appendix A.7 and A.15**, this background knowledge will serve as supplementary information about the images to assist the model in reasoning during co-testing and thus constitutes textual input alongside images. Under these settings, the dataset indeed satisfies the interleaved image-text property you mentioned.
> >
> > ---
> >
> > **R4.** We cite existing datasets only as examples of how prior work has handled non-commercial and publicly available data. Our intention is not to imply that their release automatically guarantees legal compliance, but rather that their collection and anonymization procedures provide useful methodological references.
> >
> > ---
> >
> > **R5.** We understand the reviewer’s concern regarding the distinction between human annotation and human evaluation. However, we respectfully clarify that, our goal is not to compare humans to models, but rather to test whether current MLLMs can meet the basic perception and reasoning standards that typical humans already satisfy, and provide a multi-level evaluation with image-invariable prompts. Accordingly, our tasks are grounded in everyday, non-expert knowledge, such as route planning, checking flight options, interpreting visual cues, or computing basic shopping costs. **These tasks were intentionally designed to be solvable by any typical human without specialized skills or training**.
> >
> > On this basis, because all ground-truth answers are provided by common human annotators themselves, the dataset construction inherently involves humans performing the same perception–reasoning tasks that would be measured in a formal human evaluation. As a result, the achievable human accuracy on this benchmark is trivially near 100% by design, and **running a dedicated human performance study would only reproduce what is already encoded in the annotation process**.

---

### Official Review · Reviewer_d4LY · 2025-10-30

**Soundness:** 3
**Presentation:** 3
**Contribution:** 3
**Rating:** 6
**Confidence:** 3

**Summary:**

This paper introduces "Lens," a new multi-level benchmark for evaluating multimodal large language models (MLLMs). The authors argue that existing benchmarks fail to assess how foundational perceptual skills (like object detection) contribute to higher-order reasoning. Lens addresses this by providing 3.4K contemporary images, each annotated for a hierarchy of tasks progressing from perception to understanding and reasoning. The paper evaluates over 15 recent MLLMs and finds that even top models struggle with the reasoning tasks, scoring below 60%. To address this, the authors also propose SMEC, a framework where an MLLM uses self-generated prompts to simulate a "panel of experts" to collaborate on complex reasoning, showing improved performance. A key finding is the confirmation of synergistic effects, where improving low-level perceptual tasks directly facilitates better performance on high-level reasoning.

**Strengths:**

1. Hierarchical Benchmark Design。 The "Lens" benchmark introduces a valuable multi-level structure (Perception, Understanding, Reasoning) where all tasks are annotated on the same set of images. This is a strength as it uniquely enables the evaluation of synergistic effects。

2. Contemporary and Relevant Dataset. The dataset is built from 3.4K contemporary images manually collected from social media, with a large portion (53%) published after January 2025. This freshness is crucial for fairly evaluating modern MLLMs and mitigating the risk of data contamination from older, widely used training sets.

3. Constructive Contribution with SMEC. Beyond just identifying a problem, the paper proposes a solution with the "Self-Driven Multi-Expert Collaborative Framework" (SMEC). This framework offers an innovative, tool-free method for enhancing MLLM reasoning by simulating a panel of experts. The positive results from SMEC add significant value to the paper.

**Weaknesses:**

1. Limited to Static Images. As acknowledged by the authors, the benchmark is confined to static images. This scope does not capture the complexities of real-world multimodal reasoning, which often involves video, temporal sequences, audio, or long-form narrative understanding.

2. Potential Impracticality of SMEC. The SMEC framework relies on an iterative, multi-step process of generating expert prompts and synthesizing answers. This implies a significant increase in computational overhead and latency at inference time, which could make the approach impractical for real-time applications. The paper demonstrates effectiveness but does not deeply analyze this efficiency trade-off.  For example, if using several small models, how about just using a larger powerful model? Under the same compute, which method is better?

**Questions:**

There is a formatting problem of the citations in the paper. To cite a paper, \citep should be used instead of \cite.

---

> ### Author Response · Authors · 2025-11-20
> **Response to Reviewer d4LY:**
>
> ### **W1. Limited to static images**
>
> **A1.** LENS is designed as a foundational static-image benchmark aimed at establishing principles for multi-level evaluation. Our focus in this work is on **real-world images** (non-synthetic, collected from 2024–2025 sources) with **human-curated multi-task annotations**, enabling reliable ground truth for fine-grained reasoning tasks.
>
> While video-based benchmarks are indeed crucial, especially for embodied intelligence, constructing a large-scale *real-world* video dataset with high-quality reasoning annotations would require substantially higher labor cost and logistical overhead. We thus position LENS as a solid starting point and we hope it provides meaningful insights for the community.
>
> ---
>
> ### **W2. Potential impracticality of SMEC**
>
> **A2.** SMEC is a test-time framework that trades additional reasoning steps for improved performance on complex reasoning tasks. It follows the principle of **test-time scaling**, which serves as an orthogonal axis to traditional **parameter scaling**.
>
> **Parameter scaling (larger models)** introduces substantial static costs:
>
> 1. Training compute increases by orders of magnitude, especially for the large models.
> 2. Inference memory requirements scale steeply. Parameter scaling often prohibitive in resource-constrained deployments.
>
> In contrast, **test-time scaling** allocates *additional reasoning compute only for difficult samples*, keeping the deployment footprint lightweight and practical.
>
> Moreover, recent work indicates that simply increasing model size does **not** guarantee deeper or more reliable reasoning, whereas increasing **inference-time compute** often yields significant improvements [1–3]. We empirically show that larger models (e.g., Qwen2.5VL-32b) also gain performance improvements when equipped with SMEC. This indicates that expert diversity and complementary reasoning—rather than sheer parameter count—drive the gains. Using a single larger model cannot provide the multi-perspective decomposition that SMEC introduces.
>
> **References**
>
> [1] *Chain-of-Thought Prompting Elicits Reasoning in Large Language Models.* NeurIPS 2022
>
> [2] *Large Language Models are Zero-Shot Reasoners.* NeurIPS 2022
>
> [3] *Least-to-Most Prompting Enables Complex Reasoning in Large Language Models.* ICLR 2023
>
> ---
>
> ### **Q1. Citation formatting**
>
> Thank you for pointing this out. In the revised version, we have replaced all instances of `\cite{}` with the correct `\citep{}`.

---

### Official Review · Reviewer_19vU · 2025-10-31

**Soundness:** 3
**Presentation:** 4
**Contribution:** 3
**Rating:** 8
**Confidence:** 4

**Summary:**

This paper presents Lens, a large-scale, multi-level benchmark for evaluating multimodal reasoning in large vision-language models (MLLMs).

Unlike prior task-specific datasets, Lens uses a unified image set - 3.4K high-resolution, non-commercially licensed social media image - paired with over 60K human-authored questions across 12 real-world scenarios. Each image supports eight tasks organized into three progressive tiers: perception, understanding, and reasoning, enabling a structured evaluation of how lower-level perceptual skills contribute to higher-level reasoning, while minimizing the effects of data distribution across task categories.

Comprehensive experiments on 15+ state-of-the-art MLLMs, including GPT-4o, Qwen2.5-VL, InternVL3, and reasoning MLLMs like QVQ-Max, and Kimi-VL, reveal strong interdependencies between low-level tasks, like perception and high level tasks like reasoning.

Lastly, the paper also introduces Self-Driven Multi-Expert Collaboration (SMEC), a novel framework in which an MLLM simulates a panel of specialized agents via self-generated, role-specific prompts. SMEC can dynamically refine or even redefine these roles as needed to address a given instruction without external supervision. The authors demonstrate that this approach outperforms established baselines such as majority voting, self-reflection, and direct prompting.

**Strengths:**

1. Hierarchical evaluation:
The benchmark’s three-tiered structure -  spanning perception, understanding, and reasoning - with a consistent data distribution across tasks, enables a more causal interpretation of how lower-level perceptual abilities influence higher-level reasoning (if any).

2. Scale of human annotations:
In terms of human-authored data, the dataset is impressively large and aligns with the current scale of contemporary multimodal benchmarks.

3. Up-to-date and well-curated dataset:
The dataset is well-defined and systematically structured, with 70% images sourced after November 2024 and 50% after January 2025. The images are high-resolution and reflect contemporary visual content, ensuring relevance to real-world scenarios beyond academic testbeds.

4. Strong empirical analysis:
One of the paper’s strongest aspects lies in its thorough analysis and experiments - particularly the dataset exploration in Fig. 4, cross-task synthetic analysis in Fig. 6, and detailed discussions in Sections 2.3, 4.3, 4.4, 4.5.

5. Interesting new SMEC framework:
The proposed Self-Driven Multi-Expert Collaboration (SMEC) is an interesting  idea where the same MLLM, prompted with diverse role-specific instructions, can collaboratively address complex reasoning tasks. It consistently outperforms direct prompting, self-reflection, and majority voting. This approach also raises an intriguing direction for future work on redundancy in self-reflection or voting setups - showing that structured role specialization may better harness the diverse knowledge and reasoning abilities embedded within a single MLLM?

6. Clarity and presentation:
The paper is clearly written and easy to follow, with well-organized appendices that are rich in detail and effectively referenced throughout the main text.

**Weaknesses:**

Weaknesses and Questions

1. Limited scenario diversity: While the dataset spans three main scenarios -  education, city, and home - it would benefit from broader coverage. For example, incorporating workplace, outdoor, or social-interaction settings could better capture real-world multimodal reasoning. Additionally, introducing samples grounded in logical, mathematical, or scientific domains, following the same hierarchical design principles, would strengthen the benchmark’s comprehensiveness.

2. Number and organization of tasks per tier: The benchmark currently features a relatively limited number of tasks within each tier. It is unclear whether this was an intentional design choice (e.g., grouping subtasks to facilitate downstream synergy analysis between task tiers) or simply a byproduct of human annotation diversity.

a) Did the authors design subtasks with cross-tier synergy in mind?
b) How do they envision scaling the benchmark to include additional domains (e.g., logic, mathematics, science)?

3. Clarifications on SMEC framework:
The Self-Driven Multi-Expert Collaboration (SMEC) framework is conceptually strong but would benefit from further detail and analysis. Section 3 and the appendix provide helpful descriptions, yet several aspects remain underspecified:

a) How exactly is the diversity metric defined and measured?
b)  Why did the authors choose 3500 samples subset, and how representative it is of the entire benchmark? Do results in Table 3 scale to the entire dataset?
c) What are the types of roles the model tends to generate, and which roles contribute most to performance gains?
d) are roles domain-centric  (e.g., geometry, culture, ethics) or task-centric roles (e.g., perception vs. reasoning)? Which is better and what scenario?
e)Could the framework be extended to multi-model setups, where different MLLMs act as specialized experts?

**Questions:**

Follow-up questions presented along with the weaknesses

**Details Of Ethics Concerns:**

Appendix section A.4 provides data privacy and copyright statement, but could be worth a double check to ensure the benchmark if released does not violate this.

---

> ### Author Response · Authors · 2025-11-20
> **Response to Reviewer 19vU:**
>
> ### **W1. Limited scenario diversity**
>
> **A1.** We apologize for the confusion. Our benchmark in fact follows a **three-level scene hierarchy** covering **12 specific location categories**, as stated in line 67 (“in 12 diverse scenarios”) and illustrated in the Sankey diagram in Fig. 4(a). Actually, each image filename contains a fixed three-digit scene code indicating its category (e.g., 101 = Street, 102 = Station). For clarity, we provide the distribution of images across all 12 scenarios here:
>
> |Scene Type|Street|Station|Airport|Restaurant| Scenic spot |Living room|
> |-|-|-|-|-|-|-|
> |Percentage|19.74%| 5.99%|11.55%|14.48%| 1.56%| 14.48%|
>
> |Scene Type|Bedroom| Bathroom | Kitchen |Playground|Classroom|Library|
> |-|-|-|-|-|-|-|
> | Percentage|4.73%|2.87%| 6.77%|1.53%| 8.65%| 7.66%|
> ---
> ### **W2. Number and organization of tasks per tier**
>
> **A2.** This design choice is intentional. During data construction, we first defined the full hierarchy of tasks and established precise annotation criteria for each task. Annotators were then trained to deliberately select complex real-world images and each selected image was required to receive annotations for all tasks. This ensures comprehensive coverage for downstream evaluation and provides the community with a richly annotated dataset.
>
> Annotators were encouraged (though not strictly required) to construct high-level reasoning questions that naturally depend on lower-level perceptual or grounding abilities, leveraging factual elements already covered in lower-level annotations. We also enforced **strict cross-task consistency**. For example, if a counting annotation mentions “one bicycle ridden by a person wearing a blue jacket,” then the corresponding Visual Grounding annotation must include both the bicycle and the rider.
>
> Regarding the extension of tasks to logic, mathematics, or science: our present work focuses on natural images and real-world applications, and thus we did not require annotators to possess specialized mathematical or scientific expertise. These domains are indeed valuable for assessing higher-level reasoning, and we plan to explore such extensions in future work.
>
> ### **W3. Clarifications on the SMEC framework**
>
> **A3. (a)**
> To ensure the quality of generated experts, we introduce a diversity-checking mechanism that uses the VLM itself to evaluate whether each newly generated expert both inherits the essential characteristics of its parent experts and maintains sufficient differentiation. This prevents redundant or overly similar experts and stabilizes multi-expert generation.
>
> **(b)**
> The 3,500 examples were sampled according to the **scenario distribution** of the original dataset to maintain practical applicability and representativeness. In the revised version, we additionally report results on the **full data** in revised Table 3. For Qwen2.5VL-32B:
> * **Direct (Full data):** 51.54
> * **SMEC (Full data, 3 iterations):** 54.66  *(+3.12)*
>
> These results confirm that SMEC consistently improves performance under both sampled and full-data evaluations.
>
> **(c & d)**
> We agree that this is an important question. Empirically, we observe that the model-generated roles are generally **highly aligned with the domain of the image and the scope of the question**. For example, prompts generated for images of classrooms, playgrounds, or libraries often relate to education, while those for streets or stations tend to relate to transportation. This appears to correlate with the knowledge required by the question and may imply that for certain reasoning tasks, domain-centric roles play a more significant role.
>
> **(e)**
> We agree that the method can be extended. However, the primary contribution of this work is the dataset and benchmark, and SMEC is specifically designed for the SKI task in our LENS benchmark. In future work, we plan to explore extensions where different MLLMs act as specialized experts and evaluate SMEC across diverse domain tasks. One direction is to incorporate task-specific or domain-specific models (e.g., 7B models or even non-VLM models) as expert modules, conceptually similar to Tool Calling.
>
> ### **Discussion for Strength5:**
> Thank you again for your comments. From my perspective, SMEC aims to enable the model to self-assess which domain knowledge is needed, then initialize the model based on that knowledge (without invoking external tools, but using role-specific prompts as context to elicit the MLLM’s own capabilities), and finally aggregate this knowledge to produce the final answer. This process demonstrates, to some extent, that the MLLM inherently possesses the knowledge required to answer the question. However, a single instruction or the question itself may not sufficiently activate this knowledge, or make the model aware that such knowledge is necessary for the answer.

---

> > ### Comment · Reviewer_19vU · 2025-11-26
> >
> > Thank you for the clarifications. I will retain my score

---

> > > ### Author Response · Authors · 2025-11-26
> > >
> > > Thank you very much for the positive feedback and prompt reply. We sincerely appreciate your valuable comments and diligent efforts.

---

### Official Review · Reviewer_SK8n · 2025-11-01

**Soundness:** 2
**Presentation:** 3
**Contribution:** 2
**Rating:** 4
**Confidence:** 4

**Summary:**

The paper presents LENS, a multi-level benchmark for evaluating multimodal large language models on perception, understanding, and reasoning tasks within a unified dataset. LENS contains over 60K human-written questions built on 3.4K real-world images, enabling analysis of interdependencies among visual reasoning levels. The authors further propose SMEC, a self-driven multi-expert collaboration framework that improves reasoning performance without external tools. Experiments on 15+ open and closed-source models reveal strong correlations between perception and reasoning performance and highlight persistent challenges in complex reasoning.

**Strengths:**

1. LENS unifies eight well-defined tasks over shared images, allowing systematic analysis of inter-level dependencies.
2. Ensures methodological transparency and ethical compliance through detailed dataset documentation and privacy handling.

**Weaknesses:**

1. The boundaries between perception, understanding, and reasoning tasks are not clearly defined, and some tasks (e.g. SRC as a reasoning task) overlap in scope.
2. Lacks qualitative or fine-grained error analysis, which limits understanding of how and why models fail across different levels.
3. Font size in Figure 6 should be improved for better readability.

**Questions:**

1. Could the authors clarify the rationale for categorizing SRC as a reasoning task, given that its scope appears to overlap with understanding-level tasks?
2. Would it be possible for the authors to include or discuss more detailed qualitative error analyses to better illustrate common failure patterns and model limitations across different task levels?

---

> ### Author Response · Authors · 2025-11-20
> **Response to Reviewer SK8n:**
>
> ### **W1 & Q1. Boundaries among perception, understanding, and reasoning tasks**
>
> **A1.** Our design principle follows a progressive capability hierarchy:
>
> * **Perception** focuses on recognizing visual elements (e.g., OD, OC, OE).
> * **Understanding** emphasizes grounded spatial/semantic alignment between image and text (e.g., RE, VG, OCR).
> * **Reasoning** requires integrating visual content with external knowledge, multi-step inference, or implicit contextual cues (e.g., SRC, SKI).
>
> We classify **SRC** as a *reasoning* task because it typically involves inferring spatial relationships under a constrained or altered viewpoint, rather than directly extracting spatial facts from the image. As shown in Fig. 7, most SRC questions impose a **hypothetical or shifted viewpoint** (e.g., “Suppose you are …”), requiring the model to mentally transform the scene before evaluating relative positions. Other SRC items ask about relations between entities described only in free-form text (e.g., orientation, distance), which cannot be answered through simple grounding or object localization.
>
> These steps require implicit spatial transformation, multi-cue integration, and geometric inference, going beyond understanding-level grounding. We therefore categorize SRC as **higher-order spatial reasoning**, rather than perception or understanding.
>
> ---
>
> ### **W2 & Q2. On the lack of qualitative or fine-grained error analysis**
>
> **A2.** We agree that qualitative, fine-grained error cases can substantially deepen insight. In the revised version, we added a dedicated subsection in **Appendix A.9** presenting representative failure cases across tasks.
>
> To make failure patterns clear, we separate the visualization into two categories:
> (1) **VQA-style tasks**, and
> (2) **Localization tasks** (including detection and visual grounding).
>
> **Perception & Understanding Tasks.**
> These tasks primarily require aligning visual content with textual queries. Current MLLMs generally achieve high accuracy here. Their failures are dominated by visual limitations such as small objects, occlusion, or resolution sensitivity.
>
> **Reasoning Tasks.**
> For reasoning-level questions, we observe two recurring patterns:
>
> 1. **Correct reasoning, incorrect perception.**
>    In many cases, the model’s inferential steps are logically sound but the initial visual extraction is wrong. For example, in a price-computation task, although the model performs the arithmetic correctly, it misreads the menu price of the beef bun, leading to a wrong total (5.5 instead of 7.5 yuan). This indicates that reasoning performance is still bottlenecked by perception, motivating agent-based or expert-collaborative approaches, to compensate for weak low-level signals. These tasks also require identifying small but crucial regions, suggesting that dynamic zooming or image exploration may be beneficial [1–4].
>
> 2. **Spatial and physical reasoning deficits.**
>    Models also struggle with real-world geometric or physical reasoning. For instance, answering whether two cyclists are more than 5m apart requires relating object size to physical scale and inferring distance, implicitly relying on depth cues or physical commonsense. Our examples show that current models remain limited in spatial grounding and physical reasoning, consistent with recent findings in the literature [5–6].
>
> We appreciate the reviewer’s suggestion. These qualitative analyses are now included in the paper and highlight future research directions for improving both perception and high-level reasoning in multimodal models.
>
> **References**
>
> [1] Wu, Penghao, and Saining Xie. *V?: Guided Visual Search as a Core Mechanism in Multimodal LLMs.* CVPR 2024
>
> [2] Zhang, Jiarui, et al. *MLLMs Know Where to Look: Training-Free Perception of Small Visual Details with Multimodal LLMs.* ICLR 2025
>
> [3] Shen, Haozhan, et al. *ZoomEye: Enhancing Multimodal LLMs with Human-Like Zooming Capabilities through Tree-Based Image Exploration.* EMNLP 2025
>
> [4] Shao, Hao, et al. *Visual CoT: Advancing multi-modal language models with a comprehensive dataset and benchmark for chain-of-thought reasoning.* NeurIPS 2024
>
> [5] Zhang, Wanyue, et al. *Why Do MLLMs Struggle with Spatial Understanding?  A systematic analysis from data to architecture.* arXiv 2025
>
> [6] Azzolini, Alisson, et al. *Cosmos-Reason1: From Physical Common Sense to Embodied Reasoning.* arXiv 2025
>
> ---
>
> ### **Q3. Font size in Figure 6**
>
> **A3.** We have enlarged the font size and adjusted the layout spacing in Figure 6 for readability.

---

> > ### Comment · Reviewer_SK8n · 2025-11-21
> >
> > Thank you for your response. Most of my concerns have been addressed, and I will raise my score.

---

> ### Author Response · Authors · 2025-11-22
>
> Thank you very much for the positive comments and quick reply.
> We authors really appreciate your valuable comments and hard efforts, and we are grateful for your willingness to update the score.

---

### Author Response · Authors · 2025-11-22
**Summary of Revisions**

We thank the reviewers for their constructive feedback. In this revised manuscript, we have substantially improved the paper by incorporating new analyses, experiments, and visualizations, all highlighted in **blue** in the updated submission. Below is a simple summary of the major revisions:

* **Comparison with existing benchmarks.** We added a concise paragraph in the introduction to more clearly articulate how our benchmark differs from and improves upon prior work.

* **Supplement to related work.** We added a subsection of Related work about agentic reasoning to highlight how SMEC differs from and improves upon existing approaches.

* **Additional visualizations and error analysis.** We added new qualitative visualizations (Figure 9), including expanded failure cases across tasks, and analyzed the underlying causes for these errors in relation to task types and model limitations.

* **Expanded evaluation of SMEC.** We strengthened the evaluation of SMEC on the SKI task by adding full-data experiments and more comprehensive comparisons, further validating the effectiveness of the method.

* **Improved figure readability.** We refined Figure 6 to enhance clarity and readability based on reviewer feedback.

* **Citation style update.** We replaced all `\cite{}` with the correct `\citep{}` to ensure consistent citation formatting throughout the paper.

We hope these revisions address the reviewers’ concerns and significantly strengthen the overall contribution of the paper. A detailed, point-by-point response to each reviewer is provided below.

---

### Author Response · Authors · 2025-11-29
**Summary of the Discussion Period**

We sincerely thank the reviewers and the AC for their time and constructive feedback throughout the discussion period. We fully understand the decision to revert the discussion back to prior to the discussion period, and we reiterate our commitment to **strictly following the ICLR Code of Conduct to help maintain the fairness of the peer-review process**. Based on this, we briefly summarize the progress made during the discussion phase for AC's reference.

Our initial scores were **2, 4, 6, and 8**. After actively engaging with reviewer concerns and providing additional experiments and clarifications, the reviewers who initially assigned 2 and 4 updated their scores to 4 and 6 (score pattern updated to **4, 6, 6, and 8 before Nov 27**). Below we summarize the key points raised by the two reviewers who shifted from negative to positive:

* **Reviewer SK8n (Score: 4 → 6).**
  The reviewer mainly requested clearer boundaries for the SRC task and more in-depth qualitative analysis of failure cases. In response, we explicitly clarified the task definition again, added comprehensive discussions of incorrect examples (**Appendix A.9**), and incorporating extensive new visualizations in the revision (**Figure 9**). These analyses also helped identify promising directions for future research.

* **Reviewer 4Zew (Score: 2 → 4).**
  The reviewer mainly raised concerns regarding about the motivation of the dataset, the motivation and applicability of the SMEC method, ethical considerations, and human performance. We further clarified the core contribution and motivation behind the dataset and SMEC, strengthened comparisons with existing benchmarks (**a concise paragraph in the introduction**), added the related work about agentic reasoning (**Appendix A.1.3**), and provided examples of prior work handling non-commercial and publicly available data, e.g. Laion-400m, Conceptual captions and COYO-700M. Moreover, we validated SMEC on an additional real-world multimodal reasoning benchmark to demonstrate broader applicability (**results on RealWorldQA benchmark**). To address the human-evaluation concern, we clarified that all tasks were intentionally designed to be solvable by typical humans without requiring specialized skills or training.

We hope this summary provides the AC with a clear account of the constructive progress made during the discussion period and the improvements incorporated into the revised manuscript.

---

### Meta-Review · Program_Chairs · 2026-01-03

**Summary:**

The paper initially receives scores of 4, 8, 6, and 2. After the rebuttal, two reviewers explicitly increase their scores, and one reviewer explicitly maintains the original score. The AC considers that the reviewers’ main concerns are well addressed in the rebuttal. Although the analysis based on the unified image set is not sufficiently deep and the framework is not fundamentally new, the benchmark is reasonably motivated. In particular, annotating each image for eight tasks has the potential to enable multi-dimensional analysis and provide new insights. Therefore, the AC recommends acceptance. The AC also requests that the authors carefully revise the paper based on the rebuttal, especially by strengthening the analysis of results to provide new insights rather than reiterating well-known conclusions.

**This paper is conditionally accepted provided the authors do the following for the camera-ready:**
[Ethics] Authors must describe steps taken to respect copyrights and terms of service of the websites scraped. They must acknowledge any legal risks associated with using or releasing this data, if any.

**Reviewer Concerns:**

The reviewers raised several concerns, with the key and common ones summarized below.

1. Reviewer SK8n notes the lack of qualitative or fine grained error analysis. The rebuttal provides a brief analysis attributing errors to visual limitations and deficiencies in spatial and physical reasoning. The AC considers these explanations to be common observations for non unified image set benchmarks. Given that each image is annotated for eight tasks, a deeper and more systematic analysis is necessary to provide meaningful insights for future research and should be included in a revised version.
2. Reviewers SK8n, 19vU, 4Zew request clarifications on several aspects, including task boundaries, scenario diversity, the SMEC framework, the diversity metric, the definition of interleaved image text data, and the human evaluation protocol. The rebuttal addresses these points, and reviewers SK8n and 19vU indicate that their concerns are resolved.
3. Reviewer d4LY aises concerns about the potential impracticality of SMEC. The rebuttal emphasizes that the framework supports natural test time scaling and is orthogonal to parameter scaling.
4. Reviewer 4Zew questions the novelty and contribution of the proposed framework. The rebuttal includes conceptual comparisons with related work. From the AC’s perspective, the framework provides some incremental gains, but the contribution of the benchmark itself also needs to be considered in evaluating the overall impact.
5. Reviewer 4Zew suggests validating the framework on additional datasets. The rebuttal adds experiments on the RealWorldQA dataset.

**Reviewer Scores:**

Reviewer SK8n and Reviewer 4Zew explicitly indicate that they slightly increase their scores, which are expected to reach 6 and 4, respectively. Reviewer 19vU explicitly states that they maintain the original score of 8. Reviewer d4LY mainly raises clarification related issues and expresses an overall positive attitude, and is likely to maintain a score of 6.

---

### Decision · Program_Chairs · 2026-01-26

**Decision:**

Accept (Poster)

**Comment:**

Conditions for acceptance have been satisfied.